# MELK is not necessary for the proliferation of basal-like breast cancer cells

Hai-Tsang Huang[1,2], Hyuk-Soo Seo[1], Tinghu Zhang[1,2], Yubao Wang[1,2], Baishan Jiang[1,2], Qing Li[1,2], Dennis L Buckley[3], Behnam Nabet[3], Justin M Roberts[3], Joshiawa Paulk[3], Shiva Dastjerdi[3], Georg E Winter[3], Hilary McLauchlan[4], Jennifer Moran[4], James E Bradner[3,5,6], Michael J Eck[1,2], Sirano Dhe-Paganon[1], Jean J Zhao[1,2]*, Nathanael S Gray[1,2]*

[1]Department of Cancer Biology, Dana-Farber Cancer Institute, Boston, United States; [2]Department of Biological Chemistry and Molecular Pharmacology, Harvard Medical School, Boston, United States; [3]Department of Medical Oncology, Dana-Farber Cancer Institute, Boston, United States; [4]MRC Protein Phosphorylation and Ubiquitylation Unit, College of Life Sciences, University of Dundee, Dundee, United Kingdom; [5]Department of Medicine, Harvard Medical School, Boston, United States; [6]Novartis Institutes for Biomedical Research, Cambridge, United States

**Abstract** Thorough preclinical target validation is essential for the success of drug discovery efforts. In this study, we combined chemical and genetic perturbants, including the development of a novel selective maternal embryonic leucine zipper kinase (MELK) inhibitor HTH-01-091, CRISPR/Cas9-mediated MELK knockout, a novel chemical-induced protein degradation strategy, RNA interference and CRISPR interference to validate MELK as a therapeutic target in basal-like breast cancers (BBC). In common culture conditions, we found that small molecule inhibition, genetic deletion, or acute depletion of MELK did not significantly affect cellular growth. This discrepancy to previous findings illuminated selectivity issues of the widely used MELK inhibitor OTSSP167, and potential off-target effects of MELK-targeting short hairpins. The different genetic and chemical tools developed here allow for the identification and validation of any causal roles MELK may play in cancer biology, which will be required to guide future MELK drug discovery efforts. Furthermore, our study provides a general framework for preclinical target validation.
DOI: https://doi.org/10.7554/eLife.26693.001

*For correspondence:
jean_zhao@dfci.harvard.edu (JJZ); nathanael_gray@dfci.harvard.edu (NSG)

ⓒ Copyright Huang et al. This article is distributed under the terms of the Creative Commons Attribution License, which permits unrestricted use and redistribution provided that the original author and source are credited.

## Introduction

Maternal embryonic leucine zipper kinase (MELK) is a serine/threonine kinase in the adenosine monophosphate-activated protein kinase (AMPK)-related kinase family, and was first identified as a maternal gene in mouse eggs and preimplantation embryos (*Heyer et al., 1997*). In the past decade, multiple studies have reported overexpression of MELK in various cancers, including breast (*Lin et al., 2007*), brain (*Marie et al., 2008*; *Nakano et al., 2008*), colorectal, lung and ovarian (*Gray et al., 2005*). Moreover, overexpression of MELK is observed in cancer stem cells (*Ganguly et al., 2014*; *Hebbard et al., 2010*; *Liu et al., 2006*), associated with undifferentiated phenotype (*Rhodes et al., 2004*), poor prognosis (*Nakano et al., 2008*; *Pickard et al., 2009*), and chemo and radioresistance (*Choi and Ku, 2011*; *Kim et al., 2015*; *Speers et al., 2016*). Despite a strong association with tumorigenesis, little is known about whether MELK plays a causal role in orchestrating these aggressive cancer phenotypes. Nonetheless, studies have shown that MELK may control cell cycle progression (*Davezac et al., 2002*; *Gray et al., 2005*) and suppress apoptotic

signals (*Lin et al., 2007*). Out of all of these potential roles, MELK is best characterized as a mitotic kinase, as its expression pattern highly correlates with other mitotic genes (*Badouel et al., 2010*; *Nakano et al., 2005*). At the protein level, MELK is hyperphosphorylated (*Badouel et al., 2010*), and reaches its maximal abundance (*Gray et al., 2005*) and kinase activity during M phase in both Xenopus embryos and human cancer cell lines (*Blot et al., 2002*; *Davezac et al., 2002*). Upon mitotic exit, MELK is actively degraded and dephosphorylated (*Badouel et al., 2010*).

Among MELK-associated cancers, breast cancer is one of particular interest. MELK-expressing cells enriched from mammary tumors of MMTV-Wnt1/MELK-GFP mice have higher tumor-initiating potential both in vitro and in vivo, which is reversed upon MELK-targeting shRNA knockdown (*Hebbard et al., 2010*). Furthermore, correlative studies have repeatedly identified MELK expression as a gene signature in breast cancer (*Komatsu et al., 2013*; *Liu et al., 2015*; *Pickard et al., 2009*). However, while some groups claimed that MELK overexpression is found in all breast cancers (in comparison to normal tissues), and demonstrated that MELK-targeting siRNA impaired growth in the luminal lines T-47D and MCF7 (*Lin et al., 2007*; *Pickard et al., 2009*), others claimed that MELK dependence is specific to basal-like breast cancer (BBC) (*Touré et al., 2016*; *Wang et al., 2014*). The latter argument was postulated based on higher expression of MELK in BBC (in comparison with other subtypes), and supported by subtype-specific toxicity using both shRNA depletion and pharmacological inhibition of MELK. Regardless of the subtype specificity, BBC, which largely overlaps with triple-negative breast cancer (TNBC), is in great need of a clinically tractable therapeutic target (*Foulkes et al., 2010*; *Rakha et al., 2008*). As patients with BBC/TNBC face dismal outcome upon chemoresistance, investigation of potential molecular targets, such as MELK, is worthy of attention.

Due to the potential oncogenic role of MELK, several groups have developed MELK inhibitors to validate its potential as a drug target. Among these, OTSSP167 is the most potent, with subnanomolar activity against MELK and broad-spectrum efficacy in in vitro and in vivo tumor models of various tissue origins (*Chung et al., 2012*); these encouraging preclinical results spurred initiation of a clinical trial of this compound for patients with solid tumors (*Ganguly et al., 2014*). However, OTSSP167 is a promiscuous kinase inhibitor, and it is unclear whether its antitumor activity stems from MELK inhibition. Subsequently, two pharmaceutical groups reported the development of MELK inhibitors with much improved selectivity. MELK-T1, the product of a fragment-based drug discovery campaign, exhibits impressive selectivity owing to an atypical carbonyl hinge binder. Based on profiling a panel of 235 kinases, only MELK ($IC_{50}$ = 37 nM) and Flt3 ($IC_{50}$ = 18 nM) were significantly inhibited (*Johnson et al., 2015*). Another effort successfully developed several low nanomolar MELK inhibitors based off of a 4-(pyrazol-4-yl)-pyridine scaffold. In particular, NVS-MELK8a ($IC_{50}$ = 2 nM) exhibited high selectivity with only 7 off-target kinases inhibited >85% when profiled against 456 kinases at 1 µM (*Touré et al., 2016*). Lastly, MRT199665, a pan-AMPK-related kinase inhibitor, showed reasonable potency against MELK ($IC_{50}$ = 29 nM) (*Clark et al., 2012*). All of these MELK inhibitors were evaluated in our study.

Here, we report the discovery of HTH-01-091, a potent and selective MELK inhibitor. We utilized HTH-01-091, published MELK inhibitors, as well as genetic knockout, genetic depletion, and chemical-genetic degradation strategies to study the consequences of loss of MELK function in BBC cell lines. Surprisingly, our results indicate that neither chemical inhibition nor genetic manipulation of MELK in breast cancer cell lines inhibited cell growth in vitro. The discrepancy between our study and previous reports prompted us to uncover selectivity issues of OTSSP167 and potential off-target effects from MELK-targeting short hairpins. The chemical and genetic tools presented in this study will be useful to further elucidate cellular functions of MELK, and represent a general strategy for thorough target validation.

## Results

### Discovery of lead scaffold and compound optimization for selective MELK inhibition

Based on biochemical (KINOME*scan*) and cellular (KiNativ) kinase profiling of our in-house library of structurally diverse ATP-competitive kinase inhibitors, we identified several chemotypes that were prioritized for further characterization and development. In particular, we focused on JW-7-25-1 (*Figure 1A*), a benzonaphthyridinone derivative that inhibited MELK pull-down by 100%

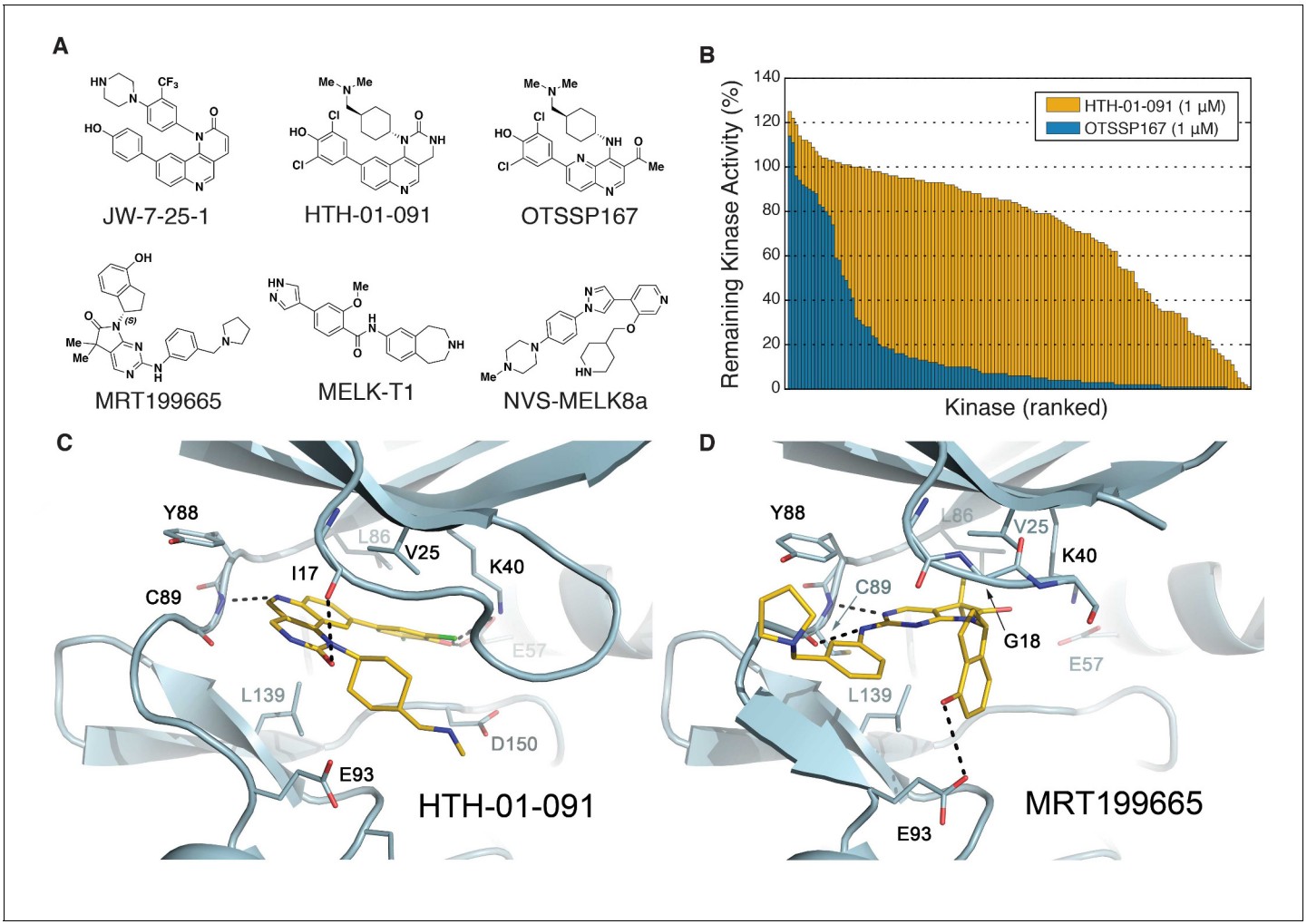

**Figure 1.** Biochemical characterization of HTH-01-091 and other MELK inhibitors. (**A**) The chemical structures of all MELK inhibitors used in this study. (**B**) Overlaid histograms comparing the percent remaining kinase activities of 140 kinases (ICKP panel) when treated with OTSSP167 (1 µM) versus HTH-01-091 (1 µM). The kinases are ranked and therefore in different orders between the two compounds. See also *Figure 1—figure supplement 1*, *Figure 1—source data 1* and *Table 2* for more inhibitor selectivity and potency data. (**C**) Crystal structure of MELK(2-333) in complex with HTH-01-091. (PDB accession code: 5TWL). (**D**) Crystal structure of MELK(1-340) in complex with MRT199665. (PDB accession code: 5TX3).

DOI: https://doi.org/10.7554/eLife.26693.002

The following source data and figure supplement are available for figure 1:

**Source data 1.** Kinase profiling of OTSSP167 and HTH-01-091 by the International Center for Kinase Profiling (ICKP).

DOI: https://doi.org/10.7554/eLife.26693.004

**Figure supplement 1.** KINOME*scan* TREE*spot* analysis of MELK inhibitors.

DOI: https://doi.org/10.7554/eLife.26693.003

(KINOME*scan*) at 10 µM. Its affinity for MELK was confirmed using the Z'-LYTE enzymatic assay ([ATP]=25 µM near Km of MELK), with a measured IC$_{50}$ value of 5.0 nM (*Table 1*).

Encouraged by the high potency of JW-7-25-1, we initiated a medicinal chemistry campaign focused on improving its kinase selectivity while preserving potency. JW-7-25-1 exhibited a broad kinase selectivity profile with an S(35) score of 0.54 when tested at 10 µM (*Figure 1—figure supplement 1*). We confirmed potent inhibition of several off-targets and were keen on reducing the inhibition of FRAP1(mTOR), PIK3CA, and CDK7, as these kinases drive proliferation in breast cancer (*Liu et al., 2009*; *Wang et al., 2015*; *Yu et al., 2001*), which would complicate the interpretation of MELK-dependent pharmacology if inhibited. Thus, we monitored inhibition of these off-target kinases throughout the optimization process. After several iterative rounds of compound synthesis and

**Table 1.** Biochemical IC$_{50}$ values of MELK inhibitors.

| | Biochemical IC$_{50}$ (nM) | | | | |
|---|---|---|---|---|---|
| | **MELK*** | **PIK3CA$^{\dagger}$** | **mTOR*** | **GSK3A*** | **CDK7$^{\dagger}$** |
| JW-7-25-1 | 5.0 | | 5.5 | 12.3 | 63.4 |
| HTH-01-091 | 10.5 | 962 | 632 | 1740 | 1230 |
| OTSSP167 | 0.5 | 66.5 | 35.7 | 1.6 | 49.1 |
| MRT199665 | 1.4–3.3 | | | | |
| NVS-MELK8a | 11.9 | | | | |
| MELK-T1 | 13.5 | | | | |

*Kinase activity measured by Z'-LYTE assay at [ATP]=apparent Km, in accordance with Z'-LYTE Screening Protocol and Assay Conditions provided by Life Technologies.

$^{\dagger}$Kinase activity measured by Adapta assay at [ATP]=apparent Km, in accordance with Adapta Screening Protocol and Assay Conditions provided by Life Technologies.

DOI: https://doi.org/10.7554/eLife.26693.005

characterization, we arrived at HTH-01-091, which demonstrated significantly improved selectivity (*Figure 1A*, *Figure 1—figure supplement 1*).

HTH-01-091 inhibits MELK with an IC$_{50}$ value of 10.5 nM, which is comparable to MELK-T1 and NVS-MELK8a in the same enzymatic assay (*Figure 1A* and *Table 1*). Furthermore, HTH-01-091 exhibited reduced inhibition of the select off-target kinases by 20- to 140-fold when compared with JW-7-25-1 (*Table 1*). Specifically, HTH-01-091 exhibited a greatly improved S(35) score of 0.16 at 1 μM (*Figure 1—figure supplement 1*).

To further investigate the selectivity profile of HTH-01-091, we used the International Center for Kinase Profiling (ICKP) panel, which measured the activity of 141 kinases using radiometric kinase assays (*Hastie et al., 2006*). Consistent with our previous result, 1 μM of HTH-01-091 selectively inhibits 4% of the kinases over 90%, whereas 1 μM of OTSSP167 inhibits 67% of the kinases over 90% (*Figure 1B* and *Figure 1—source data 1*). HTH-01-091 (1 μM) inhibited seven kinases more strongly than MELK in the ICKP panel, which includes PIM1/2/3, RIPK2, DYRK3, smMLCK and CLK2. Using the radioactive filter-binding assay provided by ICKP (*Hastie et al., 2006*), we found that HTH-01-091 inhibited MELK activity with an IC$_{50}$ value of 15.3 nM, in good agreement with the Z'-LYTE assay, while those potential off-target kinases exhibited IC$_{50}$ values in the range of 42–109 nM (*Table 2*). In sum, HTH-01-091, a potent MELK inhibitor with a greatly improved selectivity profile relative to the parent compound JW-7-25-1 as well as OTSSP167, is a valuable tool compound to interrogate MELK-dependent pharmacology.

**Table 2.** Enzymatic IC$_{50}$ values of the main targets of HTH-01-091 measured by the International Center for Kinase Profiling (ICKP).

| Kinase | % activity remaining (1 μM) | Enzymatic IC$_{50}$* (nM) |
|---|---|---|
| PIM3 | 1 | |
| PIM1 | 1 | 60.6 |
| PIM2 | 2 | |
| RIPK2 | 3 | 42.5 |
| DYRK3 | 5 | 41.8 |
| SmMLCK | 8 | 108.6 |
| CLK2 | 11 | |
| MELK | 13 | 15.3 |
| HIPK2 | 13 | |
| DAPK1 | 15 | |

*Radioactive filter-binding assay provided by the ICKP.

DOI: https://doi.org/10.7554/eLife.26693.006

## Crystal structures of HTH-01-091 and MRT199665 with MELK kinase domain

To better understand the mode of inhibitor binding, we determined crystal structures of MELK in complex with HTH-01-091 and MRT199665. Crystal structures of MELK bound to other inhibitors have been previously reported (*Cho et al., 2014*; *Johnson et al., 2015*; *Touré et al., 2016*). The 2.5 Å structure of MELK in complex with HTH-01-091 revealed a type I binding mode (*Figure 1C*). The quinoline nitrogen in the tricylic core forms the only hinge interaction with the backbone NH of Cys89. In the deep pocket, the 3,5-dichloro-4-hydroxyphenyl group engages with the catalytic (Lys40) lysine and recruits Glu57 on the αC helix. Toward the solvent, the cyclic urea group forms a hydrogen bond with Ile17 in the P-loop. In addition, the cyclohexyl group of HTH-01-091 is sandwiched between Val25 and Leu139 in a nearly orthogonal orientation with respect to the core, while projecting the (dimethylamino)methyl tail into the negatively-charged catalytic pocket made-up by Glu93, Asp150, and Glu136 (*Figure 1C*). Many of the corresponding interactions were previously observed in the MELK-OTSSP167 co-crystal structure (*Cho et al., 2014*). However, in the OTSSP167 structure, the cyclohexyl group occupies the active site in a manner that makes greater van der Waals interactions with the P-loop, causing the N-lobe to be in a slightly more closed conformation reflected by the proximity of Phe22 to the carboxyl end of the αC helix. This could contribute to the higher binding affinity of OTSSP167 to MELK.

The MRT199665 structure also revealed a type I binding mode (*Figure 1D*). The pyrimidine and aniline nitrogen atoms from the core form dual hydrogen bonds with the backbone NH and carbonyl group of Cys89 in the hinge. Toward the deep pocket, MRT199665 hydrogen bonds with Lys40, while the dimethyl group abuts the gatekeeper L86, promoting a relatively open conformation of the N-lobe. The hydroxy group of the hydroxydihydroindene group forms a hydrogen bond with Glu93, restricting the plane of the indene orthogonoal to the rest of the molecule (*Figure 1D*). The indene group is in van der Waals contact with Gly18 and Val25, but intervening residues in the P-loop are largely disordered. These co-crystal structures provide insights on how to further optimize the inhibitors.

## HTH-01-091 is cell permeable and induces MELK degradation

After confirming HTH-01-091 as a potent MELK inhibitor in biochemical assays, we next investigated whether HTH-01-091 engages MELK in cells. We performed a KiNativ assay to measure HTH-01-091 engagement with the ATP binding pocket of MELK, which would prevent active-site labeling from a lysine-reactive ATP-biotin probe (*Patricelli et al., 2007*). HTH-01-091 dose-dependently decreased MELK pull-down by streptavidin beads, demonstrating that the compound is cell permeable and binds to MELK in an ATP-competitive fashion. As a control, ERK1/2 pull-down was not affected by HTH-01-091 treatment, consistent with our profiling data showing no binding affinity of HTH-01-091 to ERK1/2 (*Figure 2A*, *Figure 1—source data 1*).

MELK-T1 was previously reported to induce cell-cycle independent degradation of MELK (*Beke et al., 2015*). Interestingly, we found that all MELK inhibitors we tested also reduced MELK protein levels in MDA-MB-468 cells, indicating that inhibitor-induced MELK degradation may be general to ATP-competitive MELK inhibitors (*Figure 2B*). Inhibitor potency generally correlated with the rate and extent of MELK downregulation. However, MRT199665 induced the most rapid degradation of MELK protein despite being less potent than OTSSP167 based on biochemical $IC_{50}$ values. Consistent with the prior MELK-T1 report, inhibitor-induced MELK degradation requires the ubiquitin-proteasome system, as pretreatment with the proteasome inhibitor carfilzomib prevented degradation (*Figure 2—figure supplement 1*) (*Beke et al., 2015*). As the molecular mechanism involved in MELK degradation has not been described, we tried to rescue MELK degradation with the neddylation inhibitor MLN4924. However, this had minimal effect, indicating that the cullin-RING E3 ligase family is not involved in MELK degradation (*Figure 2—figure supplement 1*).

## Selective MELK inhibitors exhibit minor antiproliferative effects in breast cancer cells

As shRNA-mediated knockdown of MELK was previously reported to inhibit the proliferation of BBC cells, we next tested whether pharmacological inhibition of MELK would have similar effects. We treated a panel of breast cancer cell lines, including MDA-MB-468 and ZR-75-1 cells, with HTH-01-

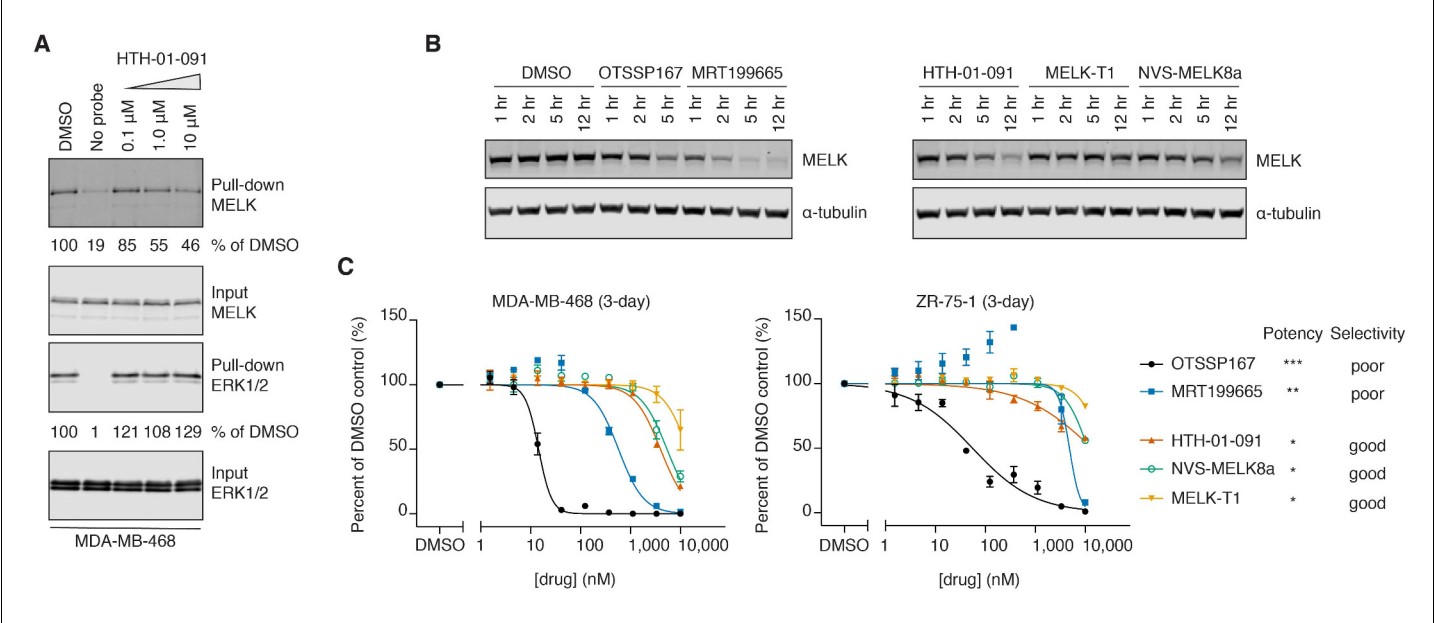

**Figure 2.** HTH-01-091 is cell permeable, causes MELK degradation, but demonstrates poor antiproliferative effects in basal-like breast cancer cell lines. (A) Immunoblots analyzing the pull-down levels of MELK and ERK1/2 by ATP-biotin probes and streptavidin beads from MDA-MB-468 cells treated with different doses of HTH-01-091 and 10 mM MG132 for 1 hour. (B) Immunoblots for MELK and α-tubulin after treatment of MDA-MB-468 cells with 1 μM of OTSSP167, MRT199665 or 10 μM of HTH-01-091, MELK-T1 and NVS-MELK8a for specified amount of time. See also *Figure 2—figure supplement 1* for rescue experiments using carfilzomib and MLN4924. (C) 3-day antiproliferation assays of HTH-01-091 and other MELK inhibitors tested in MDA-MB-468 (basal) and ZR-75-1 (luminal) cells. Values represent two independent experimental mean ±SD. Error bars shorter than the height of the symbol are not drawn. Potency (MELK): ***$IC_{50}$ <1 nM, **$IC_{50}$ <5 nM, *$IC_{50}$ <20 nM, measured by Z'LYTE biochemical assay. See also *Tables 3* and *4* for a summary of antiproliferative $IC_{50}$ values in a panel of breast cancer cell lines at 3 and 7 days post treatment, respectively.

DOI: https://doi.org/10.7554/eLife.26693.007

The following figure supplement is available for figure 2:

**Figure supplement 1.** MELK inhibitor-induced MELK degradation is dependent on the proteosome, but not the cullin-RING ubiquitin ligase family.

DOI: https://doi.org/10.7554/eLife.26693.008

091 and other MELK inhibitors; despite our focus on basal-like breast cancers, we included luminal cell lines for comparison. In 3-day proliferation assays, OTSSP167 demonstrated exceptional antiproliferative effects, exhibiting half maximal killing below 100 nM for most cell lines tested (*Figure 2C* and *Table 3*). In contrast, HTH-01-091 and the other selective MELK inhibitors exhibited antiproliferative $IC_{50}$ values that were at least 300-fold less potent. MRT199665, which exhibited intermediate potency as a biochemical MELK inhibitor, also exhibited intermediate growth inhibition in most cell

**Table 3.** 3-day antiproliferative activities of MELK inhibitors in a panel of breast cancer cell lines*.

| | 3-day antiproliferative $IC_{50}$ (μM) | | | | | | Note | |
| --- | --- | --- | --- | --- | --- | --- | --- | --- |
| Subtype | Basal-like | | | Luminal | | | | |
| Cell line | MDA-MB-468 | BT-549 | HCC70 | ZR-75-1 | MCF7 | T-47D | MELK $IC_{50}$ (nM) | Selectivity |
| HTH-01-091 | 4.00 | 6.16 | 8.80 | >10 | 8.75 | 3.87 | 10.5 | Good |
| OTSSP167 | 0.014 | 0.021 | 0.034 | 0.055 | 0.035 | 0.106 | 0.5 | Poor |
| MRT199665 | 0.58 | 0.40 | 0.39 | 4.83 | 0.44 | 5.89 | 1.4–3.3 | Poor |
| NVS-MELK8a | 5.41 | 8.05 | 5.99 | >10 | 6.06 | >10 | 11.9 | Good |
| MELK-T1 | >10 | | | >10 | | | 13.5 | Good |

*$IC_{50}$ values were estimated based on 'log(inhibitor) vs. normalized response – Variable slope' using GraphPad Prism 7. Experiments were performed in duplicates.

DOI: https://doi.org/10.7554/eLife.26693.009

lines (*Figure 2C* and *Table 3*). We evaluated the inhibitors in longer-term (7-day) proliferation assays and still observed the same trend, and no clear difference between the breast cancer subtypes (*Table 4*). We authenticated our MDA-MB-468 line by short tandem repeat analysis, and repeated the proliferation assays using MDA-MB-468 cells from a second source to confirm the modest antiproliferative effects of the selective inhibitors (*Table 4*). In brief, the selective MELK inhibitors exhibited weak antiproliferative activities against both basal-like and luminal breast cancer cell lines in vitro, casting doubt on whether MELK inhibition affects viability. However, as the non-selective OTSSP167 and MRT199665 are also more potent inhibitors of MELK, these data alone were not sufficient to allow us to deduce whether their stronger antiproliferative potency stems from on- or off-target effects.

## MELK−/− MDA-MB-468 cells exhibit normal growth and are not less sensitive to MELK inhibitors

The difference in antiproliferative effect between OTSSP167 and HTH-01-091 suggested that MELK inhibition might not be the primary contributor to the cellular activity of OTSSP167. To address this question, we used CRISPR/Cas9 to genetically delete MELK in MDA-MB-468 cells. We designed four independent guide sequences targeting exon 4 or 5 of *MELK*, which are situated in the kinase domain. We selected sgMELK-3 for further assessment due to high efficiency of indels generated and effective loss of MELK protein levels (*Figure 3—figure supplement 1A,B*). Notably, sgMELK-3 presents desirable target specificity, as there are no sequences within the human genome with 3 or less mismatches based on ZiFit Targeter (*Sander et al., 2010*). We proceeded with single cell cloning, and out of the 10 clones isolated, 6 clones were completely deficient of MELK protein, as indicated by the use of a rabbit monoclonal antibody raised against an amino-terminal epitope of MELK (EPR3981, Abcam or GeneTex) (*Figure 3—figure supplement 1C*). Genotyping analysis around the cut site aided by TIDE (*Brinkman et al., 2014*) further confirmed frame shifts or larger deletions on both alleles (*Figure 3—figure supplement 2*, *Figure 3—source data 1*).

We compared a WT clone (clone E9) and a MELK−/− clone (clone C7) with the parental cell line, and observed no significant difference in growth rate or cell cycle distribution (*Figure 3A,B,C*). We also compared the sensitivity of the WT and the MELK−/− cells to OTSSP167 and the selective MELK inhibitors in 3-day proliferation assays (*Figure 3D*). If the antiproliferative activity of OTSSP167 was primarily due to MELK inhibition, then we would expect MELK−/− cells to exhibit reduced sensitivity to OTSSP167 treatment. However, we did not observe differences in cell viability after treatment with OTSSP167 or any of the more selective MELK inhibitors, indicating that the activity of MELK inhibitors in MDA-MB-468 cells was driven by inhibition of targets other than MELK.

## dTAG-mediated loss of MELK does not impair growth of MDA-MB-468 cells

As the process for deriving and isolating clonal lines of MELK−/− MDA-MB-468 cells requires an extended period of time, we were concerned that these clonal lines would be able to compensate for loss of MELK during this process. Thus, to understand the immediate effect of MELK loss, we

**Table 4.** 7-day antiproliferative activities of MELK inhibitors in a panel of breast cancer cell lines*.

| | 7-day antiproliferative IC$_{50}$ (μM) | | | | | |
| --- | --- | --- | --- | --- | --- | --- |
| | **Basal-like** | | | | **Luminal** | |
| Cell line | MDA-MB-468 | MDA-MB-468 (second source) | BT-549 | HCC70 | MCF7 | T-47D |
| HTH-01-091 | 2.71 | 10.7 | 2.82 | 2.43 | 4.13 | 0.78 |
| OTSSP167 | 0.012 | 0.033 | 0.009 | 0.021 | 0.027 | 0.009 |
| MRT199665 | 0.16 | 0.91 | 0.31 | 0.075 | 0.11 | 0.62 |
| NVS-MELK8a | 2.96 | 8.4 | 4.98 | 4.17 | 2.81 | 4.90 |

*IC$_{50}$ values were estimated based on 'log(inhibitor) vs. normalized response – Variable slope' using GraphPad Prism 7. Experiments were performed in duplicates.

DOI: https://doi.org/10.7554/eLife.26693.010

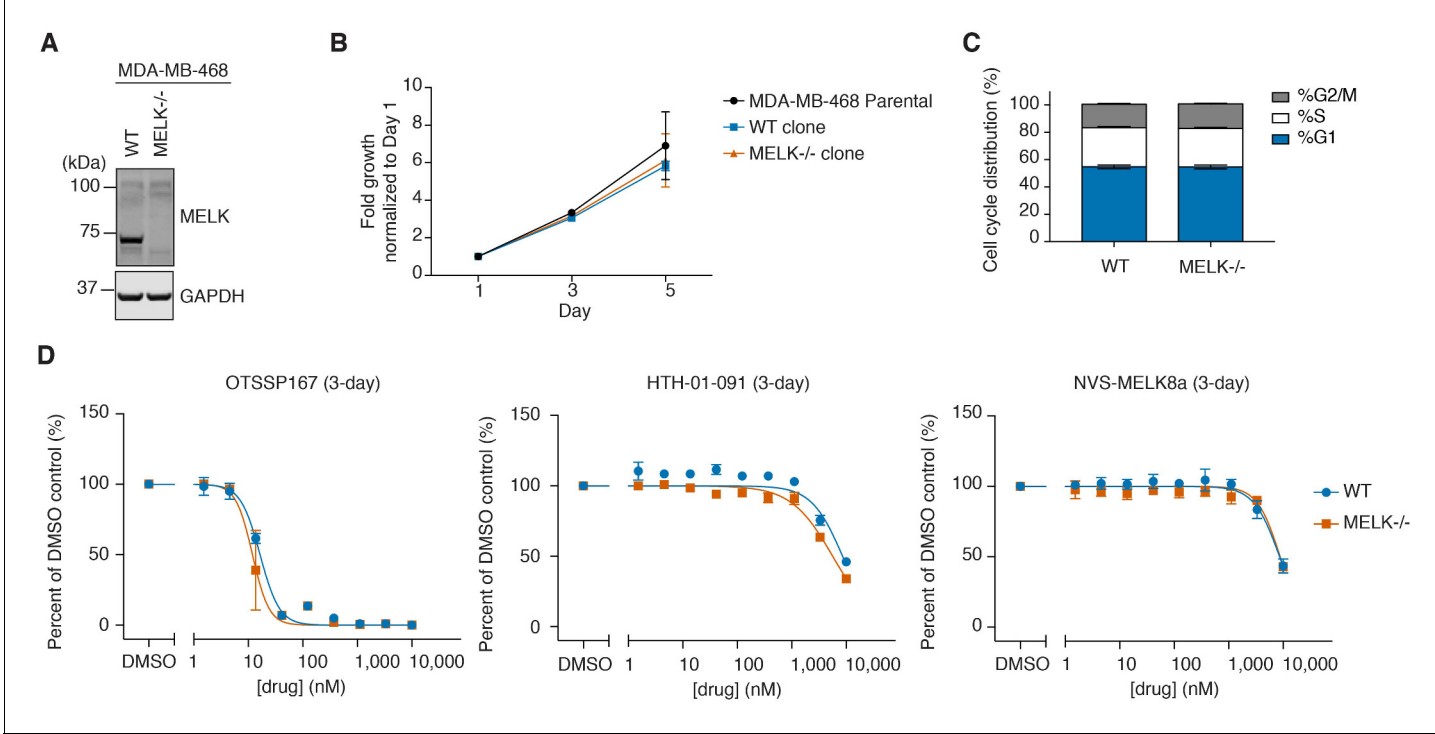

**Figure 3.** WT and MELK−/− MDA-MB-468 cells have similar growth rates, cell cycle distribution, and sensitivity to MELK inhibitors. (**A**) Immunoblots for MELK and GAPDH in WT (clone E9) and MELK−/− (clone C7) clones isolated from MDA-MB-468 cells transfected with sgMELK-3 and Cas9. See also *Figure 3—figure supplements 1* and *2*, and *Figure 3—source data 1* for the details of the generation of MELK−/− clones. (**B**) 5-day proliferation curves comparing the growth rates of parental, WT, and MELK−/− MDA-MB-468 cells. Values represent two independent experimental mean ±SD. (**C**) Cell cycle analysis comparing WT and MELK−/− MDA-MB-468 cells. Values represent triplicate mean ±SD. (**D**) 3-day proliferation assays comparing the sensitivity of WT and MELK−/− MDA-MB-468 cells to OTSSP167, HTH-01-091 and NVS-MELK8a. Values represent duplicate mean ±SD. Error bars shorter than the height of the symbol are not drawn.

DOI: https://doi.org/10.7554/eLife.26693.011

The following source data and figure supplements are available for figure 3:

**Source data 1.** Genotype analysis of single cell clones selected from MDA-MB-468 cells transfected with Cas9/sgMELK-3.
DOI: https://doi.org/10.7554/eLife.26693.014

**Figure supplement 1.** CRISPR/Cas9-mediated knockout of MELK in MDA-MB-468 cells.
DOI: https://doi.org/10.7554/eLife.26693.012

**Figure supplement 2.** Genotype analysis of cell clones selected from MDA-MB-468 cells transfected with Cas9/sgMELK-3.
DOI: https://doi.org/10.7554/eLife.26693.013

employed a novel chemical genetic system (the dTAG system) whereby tagged proteins are targeted for degradation by the E3 ubiquitin ligase cereblon (CRL4$^{CRBN}$) (*Erb et al., 2017*). In this system, mutant FKBP12 (FKBP12$^{F36V}$) serves as a degradation tag (dTAG) and is fused to a protein of interest. The F36V mutation introduces a 'hole' in the FKBP12 binding site that accommodates a 'bump' on the FKBP12$^{F36V}$-binding ligand, which does not effectively bind to wild-type FKBP12 (*Clackson et al., 1998*). We synthesized heterobifunctional molecules (dTAG molecules) by conjugating FKBP12$^{F36V}$ binders to thalidomide, which is a potent ligand for CRL4$^{CRBN}$. These molecules bring the FKBP12$^{F36V}$-fusion protein and CRL4$^{CRBN}$ into close proximity, thus inducing rapid ubiquitination and subsequent proteasomal degradation of the tagged protein while sparing endogenous FKBP12 (*Erb et al., 2017*; *Winter et al., 2015*).

To maintain continuous expression of MELK, we first expressed N-terminally tagged MELK (FKBP12$^{F36V}$-MELK) in MDA-MB-468 cells, and then deleted endogenous MELK using CRISPR/Cas9. A single point mutation in the protospacer adjacent motif targeted by sgMELK-3 (termed sg3R) prevented CRISPR editing of the transgene encoding FKBP12$^{F36V}$-MELK(sg3R). We isolated 24 clones with varying levels of FKBP12$^{F36V}$-MELK(sg3R) expression and varying endogenous MELK status

(*Figure 4—figure supplement 1*). Two validated MELK−/− clones expressing high levels of FKBP12$^{F36V}$-MELK(sg3R) were chosen for further studies. Importantly, the exogenous MELK fusion protein was still sensitive to MRT199665-induced degradation, and was stabilized and hyperphosphorylated during mitosis, suggesting that FKBP12$^{F36V}$-MELK(sg3R) is similarly regulated as endogenous MELK (*Figure 4—figure supplement 2*).

Four dTAG molecules (7, 13, 36 and 47) that vary in linker length and chemical structure were tested for their efficiency at depleting FKBP12$^{F36V}$-MELK(sg3R) (*Figure 4A*, *Figure 4—figure supplement 3*). All four degraders efficiently depleted FKBP12$^{F36V}$-MELK(sg3R) within 4 hours (*Figure 4B*); in particular, dTAG-13, 36, and 47 demonstrated sustained degradation of FKBP12$^{F36V}$-MELK(sg3R) for up to 72 hours (*Figure 4C*). A multiplexed quantitative mass spectrometry-based proteomics experiment demonstrated that only FKBP12$^{F36V}$-MELK was significantly degraded, confirming the selectivity of the system (*Figure 4D*) (*McAlister et al., 2012*). In a 9-day proliferation assay, neither of the FKBP12$^{F36V}$-MELK(sg3R) MELK−/− clones exhibited growth impairment when treated by dTAG-47 (*Figure 4E*), confirming that MDA-MB-468 cells are not sensitive to acute and sustained loss of MELK in vitro.

## Potential off-target activities of MELK-targeting shRNAs may explain the previous discovery of MELK dependency in BBC

The lack of antiproliferative response to selective MELK inhibition, genetic deletion, and chemical-induced degradation contradicted previous findings, where MELK-targeting short hairpins impaired BBC proliferation (*Touré et al., 2016*; *Wang et al., 2014*). To reconcile this discrepancy, we repeated the proliferation assay, using the same doxycycline-inducible shRNAs from previous reports to knockdown MELK in MDA-MB-468 cells. Indeed, shMELK-1 and shMELK-2 efficiently downregulated MELK mRNA and protein levels and impaired the growth of MDA-MB-468 cells (*Figure 5A,B* and *Figure 5—figure supplement 1A*).

Since off-target activity in RNAi technology is a known issue that is difficult to predict based on target sequences (*Jackson and Linsley, 2010*), we decided to test 3 additional MELK-targeting shRNAs selected from the RNAi Consortium (TRC) shRNA library—named shMELK-3, 4 and 5. Two days after induction, shMELK-5 resulted in similar levels of MELK knockdown compared with shMELK-1 and 2, while shMELK-3 and 4 induced only about 50% MELK knockdown (*Figure 5—figure supplement 1A,B*). Similar to previous studies using the same short-hairpin sequence (*Touré et al., 2016*), shMELK-5 induced cell growth defects, even though the effect was milder than shMELK-1 and 2 (*Figure 5B*). Interestingly, for the less efficient hairpins shMELK-3 and 4, shMELK-4 also impaired the growth of MDA-MB-468 cells while shMELK-3 did not, further demonstrating that different shRNAs can lead to discrepant phenotypic outcomes (*Figure 5—figure supplement 1C*). We observed sustained knockdown of MELK by shMELK-3 and 5 after an 8-day treatment, confirming that the growing cells maintained MELK knockdown (*Figure 5—figure supplement 1D*).

To test out whether off-targets contribute to the antiproliferative effects of shMELK-1, 2, and 5, we studied how these hairpins would affect the growth of MELK−/− MDA-MB-468 cells. Indeed, the MELK-targeting hairpins caused almost identical proliferation phenotypes in a MELK−/− background (*Figure 5C,D*, and *Figure 5—figure supplement 1E*), suggesting that off-target effects may contribute to the observed growth defects.

## Doxycycline-inducible MELK knockdown mediated by CRISPR interference does not cause proliferative defects in MDA-MB-468 cells

Even though we demonstrated that off-targets may contribute to the antiproliferative effect of the MELK-targeting shRNAs, we were surprised that three independent hairpins all impeded the growth of MDA-MB-468 cells. To improve our understanding of the phenotypic outcomes of MELK knockdown, we decided to apply CRISPR interference, an orthogonal and potentially more specific method than RNA interference to downregulate gene expression (*Gilbert et al., 2014*).

We first transduced MDA-MB-468 cells, and via flow cytometry-based cell sorting obtained cells expressing KRAB-dCas9, a CRISPR-mediated transcriptional repressor (*Gilbert et al., 2014*). We tested 8 guide sequences that target the region around the transcriptional start site of the *MELK* gene, and confirmed that 7 out of 8 guide sequences efficiently suppressed MELK transcript levels in MDA-MB-468-KRAB-dCas9 cells (*Figure 5E* and *Figure 5—figure supplement 2A*). To understand

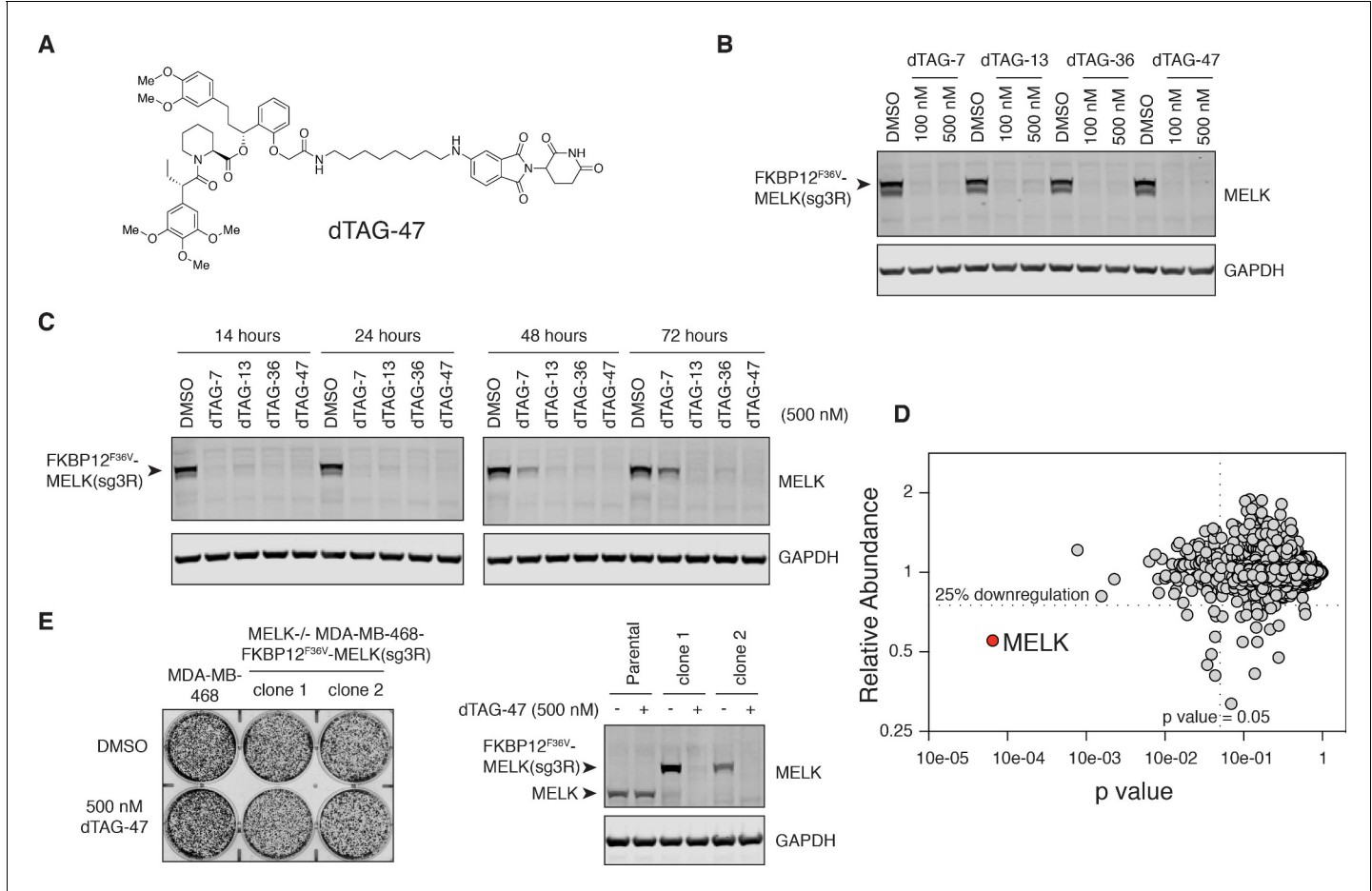

**Figure 4.** MELK−/− MDA-MB-468-FKBP12$^{F36V}$-MELK(sg3R) cells grow normally in response to pharmacologically induced FKBP12$^{F36V}$-MELK degradation. (**A**) Chemical structure of heterobifunctional dTAG molecule dTAG-47. See also *Figure 4—figure supplement 3* for the chemical structures of dTAG-7, dTAG-13 and dTAG-36. (**B**) Immunoblots for MELK and GAPDH after treatment of dTAG-7, 13, 36, and 47 at indicated concentrations in MELK−/− MDA-MB-468-FKBP12$^{F36V}$-MELK(sg3R) cells for 4 hours. See *Figure 4—figure supplement 1* for details of the generation of MELK−/− MDA-MB-468-FKBP12$^{F36V}$-MELK(sg3R) clones. (**C**) Same as in 4B, but with treatment at 500 nM and extended treatment times for 14, 24, 48, and 72 hours. (**D**) Changes in abundance of 7270 proteins (peptide count ≥2) comparing MELK−/− MDA-MB-468-FKBP12$^{F36V}$-MELK cells treated with dTAG-7 (250 nM) or DMSO for 1 hour, versus p-value (dTAG-7: triplicate, DMSO: duplicate, *limma* moderated t-test). See also *Figure 4—source datas 1* and *2* for the original and the processed data. (**E**) Crystal violet staining image showing parental MDA-MB-468 cells and MELK−/− MDA-MB-468-FKBP12$^{F36V}$-MELK(sg3R) cells after treatment with DMSO or 500 nM of dTAG-47 for 9 days. Immunoblots showing MELK and GAPDH from a duplicate plate on Day 9 confirmed sustained depletion of FKBP12$^{F36V}$-MELK(sg3R). These are representative data from one of two independent experiments.

DOI: https://doi.org/10.7554/eLife.26693.015

The following source data and figure supplements are available for figure 4:

**Source data 1.** Raw integrated ion intensities from quantitative multiplexed proteomics.
DOI: https://doi.org/10.7554/eLife.26693.019

**Source data 2.** Processed Data and Statistical Analysis by *limma* (topTable).
DOI: https://doi.org/10.7554/eLife.26693.020

**Figure supplement 1.** Single clones selected from MDA-MB-468-FKBP12$^{F36V}$-MELK(sg3R) cells subject to CRISPR/Cas9-mediated knockout of endogenous MELK.
DOI: https://doi.org/10.7554/eLife.26693.016

**Figure supplement 2.** FKBP12$^{F36V}$-MELK(sg3R) recapitulates two phenotypes of endogenous MELK to suggest preserved functions.
DOI: https://doi.org/10.7554/eLife.26693.017

**Figure supplement 3.** Chemical structures of dTAG molecules.
DOI: https://doi.org/10.7554/eLife.26693.018

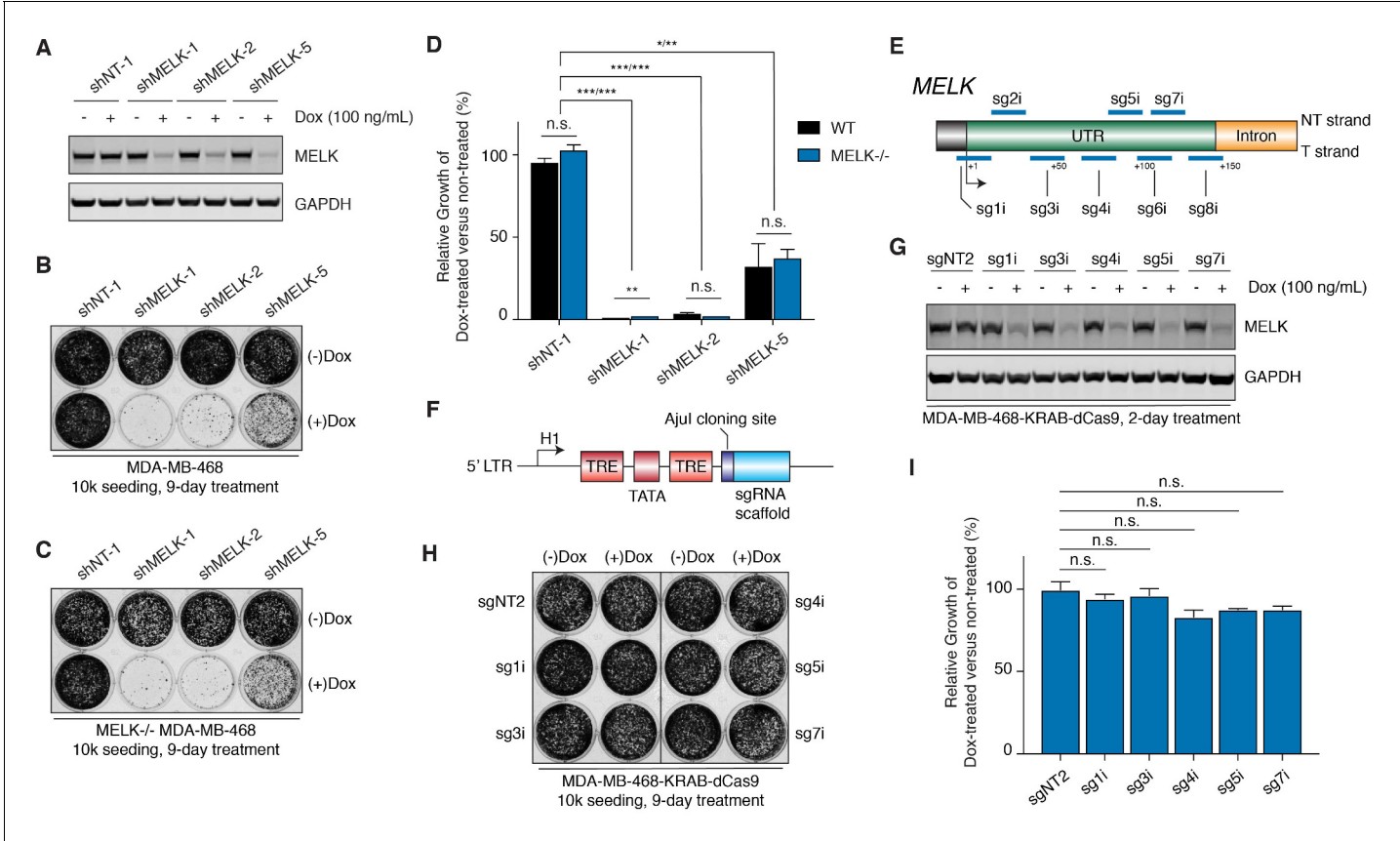

**Figure 5.** MELK-targeting shRNAs may induce antiproliferation of MDA-MB-468 cells through off-target activities while CRISPRi-mediated MELK knockdown does not affect proliferation. (**A**) Immunoblots for MELK and GAPDH after 2-day treatment with or without doxycycline (100 ng/mL) in MDA-MB-468 cells transduced with doxycycline-inducible shRNA constructs. NT-1 represents a non-targeting control. Hairpins shMELK-1, 2, and 5 target three different regions within MELK's coding region and 3'-UTR. See also *Figure 5—figure supplement 1*. (**B**) A crystal violet staining image of MDA-MB-468 cells transduced with doxycycline-inducible shRNA constructs after treatment with or without doxycycline (100 ng/mL) for 9 days with an initial seeding of 10,000 cells in a 12-well plate. (**C**) A crystal violet staining image of MELK−/− MDA-MB-468 cells (clone C7) transduced with doxycycline-inducible shRNA constructs after treatment with or without doxycycline (100 ng/mL) for 9 days with an initial seeding of 10,000 cells in a 12-well plate. See also *Figure 5—figure supplement 1E*. (**D**) A bar graph quantifying the intensity of crystal violet staining of cells treated as described in B and C. Values from the doxycycline treated groups are normalized to the untreated. Values represent mean ±SD from two independent experiments (*p≤0.05; **p≤0.01; ***p≤0.001, two-tailed Student's t-Test). (**E**) A scheme that depicts the guide sequences that target the transcription start site of *MELK* and were tested in our study. See also *Figure 5—figure supplement 2A*. (**F**) An H1-based doxycycline-inducible sgRNA construct was modified from tet-pLKO-puro, which constitutively coexpresses tetR (tet repressor protein). TRE, tet response element. (**G**) Immunoblots for MELK and GAPDH after 2-day treatment with or without doxycycline (100 ng/mL) in MDA-MB-468-KRAB-dCas9 cells transduced with doxycycline-inducible sgRNA constructs. See also *Figure 5—figure supplement 2B*. (**H**) A crystal violet staining image of MDA-MB-468-KRAB-dCas9 cells transduced with doxycycline-inducible sgRNA constructs after treatment with or without doxycycline (100 ng/mL) for 9 days with an initial seeding of 10,000 cells in a 12-well plate. See also *Figure 5—figure supplement 2C*. (**I**) A bar graph quantifying the intensity of crystal violet staining of cells treated as described in H. Values from the doxycycline treated groups are normalized to the untreated. Values represent mean ±SD from two independent experiments (for n.s., p>0.05, two-tailed Student's t-Test).

DOI: https://doi.org/10.7554/eLife.26693.021

The following figure supplements are available for figure 5:

**Figure supplement 1.** Validating MELK-targeting short hairpins.
DOI: https://doi.org/10.7554/eLife.26693.022
**Figure supplement 2.** Validating MELK knockdown mediated by CRISPR interference.
DOI: https://doi.org/10.7554/eLife.26693.023

the immediate response of MELK knockdown and to design experiments comparable to our shRNAs, we cloned the 5 most efficient guide sequences into a modified doxycycline-inducible shRNA vector (tet-pLKO-puro) (*Wiederschain et al., 2009*) where the region encoding for shRNA is replaced with an AjuI cloning site followed by the sgRNA scaffold (*Figure 5F*). Doxycycline treatment of MDA-MB-468 cells that stably express KRAB-dCas9 and the doxycycline-inducible sgRNA constructs caused efficient MELK knockdown, which was comparable to the MELK-targeting shRNAs (*Figure 5G* and *Figure 5—figure supplement 2B*). In a 9-day proliferation assay, we did not observe a significant difference between doxycycline-treated versus non-treated groups for all five sgRNAs, suggesting that MELK expression is not required for the fitness of MDA-MB-468 cells (*Figure 5H,I*, and *Figure 5—figure supplement 2C*).

## Discussion

The dependence on MELK for survival in basal-like breast cancers was previously demonstrated by MELK knockdown using shRNA in both in vitro and in vivo models (*Touré et al., 2016*; *Wang et al., 2014*). As there is still no tractable target identified in BBC, the finding encouraged a medicinal chemistry campaign to validate the therapeutic potential of MELK inhibition. However, the highly discrepant antiproliferative effects observed between the selective MELK inhibitor HTH-01-091 and the clinical candidate OTSSP167 led us to reexamine whether MELK is necessary for the survival of BBC. To answer this question, we applied and integrated multiple chemical and genetic tools, including selective MELK inhibitors, CRISPR gene editing, a chemical-induced degradation strategy (the dTAG system), RNA interference and CRISPR interference, to understand how a BBC cell line responds to loss of MELK function. Collectively, our efforts led to the conclusion that inhibition or depletion of MELK alone does not impair the proliferation of BBC cell lines in common culture conditions.

While numerous methods are available for assessing kinase inhibitor selectivity, the potential for additional unexpected off-targets can never be excluded. In addition to HTH-01-091, which exhibits substantially improved kinome selectivity in comparison with OTSSP167, we included MRT199665, NVS-MELK8a and MELK-T1 when we surveyed the proliferative response of a panel of breast cancer cell lines to MELK inhibition. Testing multiple inhibitors derived from diverse chemical scaffolds decreases the chances of chemically perturbing a common off-target, bolstering the robustness of the conclusion drawn. When we observed that three selective MELK inhibitors all showed much poorer antiproliferative effects than OTSSP167, which we recognized as multi-targeted by kinome profiling, we suspected OTSSP167 achieved its effect as a result of polypharmacology. Until recently, little had been done to validate whether the anticancer activity of OTSSP167 originated from MELK inhibition. A study investigating the abrogation of mitotic checkpoint by OTSSP167 illustrated a specific example where inhibition of several mitotic kinases other than MELK contributed to the phenotype (*Ji et al., 2016*). In addition, a CRISPR/Cas9-focused study that reached similar conclusions to our study, demonstrated that off-target mechanisms contribute to the anticancer effects of OTSSP167 because WT and MELK−/− cancer cell lines were similarly sensitive to OTSSP167 treatment (*Lin et al., 2017*). Similarly, we found that off-targets also contribute to the weak antiproliferative activities of HTH-01-091 and NVS-MELK8a, underscoring the importance of using genetic methods to examine the outcomes of chemical perturbations. The lack of strong antiproliferative activities of NVS-MELK8a in MDA-MB-468 cells contradicted a previous report (*Touré et al., 2016*), which remains to be understood; cell seeding density and compound renewals are potential reasons for the discrepant antiproliferative responses. In addition, we cannot rule out the possibility that MELK inhibition contributes to the efficacy of OTSSP167. While equivalent levels of cell killing by OTSSP167 in a MELK wild-type and a MELK-null background confirmed the presence of off-target effects, an OTSSP167-resistant MELK mutant would be necessary to fully dissect the role of MELK inhibition, which is the gold standard for validating on-target drug action (*Kaelin, 2017*). While OTSSP167 may achieve therapeutic effects through unknown mechanisms, in which MELK may or may not be involved, our data highly discourage using OTSSP167 as a probe to elucidate MELK-dependent pharmacological effects.

Our ability to isolate MELK-deficient MDA-MB-468 clones edited by the CRISPR/Cas9 system also suggested that MELK might not be necessary for in vitro proliferation. It is possible that alternate pathways readily compensate for the loss of MELK, or that resistance emerged. We inferred that the

former case is more likely based on the probability of observing 6 out of 10 isolated clones with MELK deletion. If MELK is a survival dependency that requires resistance events to overcome, we would expect a disproportionate excess of WT MELK clones after the single-cell cloning process.

However, to truly understand the transient and adaptive response to loss of MELK, we adopted a novel chemical-induced protein degradation strategy known as the dTAG system, which allows rapid control of MELK protein levels. By introducing exogenous FKBP12$^{F36V}$-MELK(sg3R) into MDA-MB-468 cells before knockout of endogenous MELK, we ensured that the cells never fully lost MELK expression. After treating these cells with the dTAG molecules to induce acute and sustained degradation of FKBP12$^{F36V}$-MELK(sg3R), we observed normal growth, which argued against an initial loss of fitness that is rescued over time. One important aspect of this strategy is to ensure that the FKBP12$^{F36V}$-fusion protein behaves similarly to the endogenous counterpart. Although we were not able to directly test the kinase function of FKBP12$^{F36V}$-MELK(sg3R) due to a lack of well-validated and specific MELK substrates, we confirmed that FKBP12$^{F36V}$-MELK(sg3R) was sensitive to MRT199665-induced degradation and was stabilized and hyperphosphorylated during mitosis, two reliable phenotypes that we and others have observed for endogenous MELK.

Using the pair of WT and MELK$-/-$ MDA-MB-468 cells, we found that off-target mechanisms may also contribute to the antiproliferative effects of shMELK-1, 2 and 5. Since the previous study was able to rescue the antiproliferative activity observed for shMELK-2 using an shRNA-resistant MELK (*Wang et al., 2014*), we postulated that the potential off-target of shMELK-2 might only manifest its effect in the presence of MELK knockdown, a so-called 'synthetic lethal' interaction. Based on the data, we encourage cautious evaluation of the phenotypes caused by RNAi-mediated MELK knockdown (*Jackson and Linsley, 2010*), and further validation of the outcomes using independent experimental approaches such as CRISPR interference. Nevertheless, it is still surprising that three independent MELK-targeting hairpins all led to loss of fitness due to off-target effects. In the future, it may be worthwhile to understand the mechanism by which these hairpins mediate growth defects, and whether it occurred by chance or in a MELK-dependent manner.

We discovered that drug-induced degradation of MELK is general to the MELK inhibitors tested. While the rate and extent of degradation largely correlated with biochemical potency for MELK inhibition, we noticed that different chemical scaffolds displayed different propensities for inducing degradation. Specifically, the pyrrolopyrimidine MRT199665 induced MELK degradation most efficiently. We postulate that the induced conformation upon binding of MRT199665 is more poised for the degradation response. The inhibitor-induced MELK degradation is unlikely cell-cycle dependent because the cells were not synchronized and we observed unchanging levels of cyclin B1 when MELK level significantly decreased (data not shown). The previous study proposed two possible models for inhibitor-induced MELK degradation: inhibitor-induced modulation of the interaction between MELK and the Hsp90-Cdc37 chaperone system, and inhibitor stabilizing an active kinase conformation that is associated with a shorter protein half-life (*Beke et al., 2015*). These two models are not mutually exclusive and may also relate to how the inhibitor-induced conformation influences auto-phosphorylation as well as phosphorylation of MELK by other kinases. The status of MELK phosphorylation has been proposed as a mechanism that regulates its stability during mitosis (*Badouel et al., 2010*).

MELK likely has complex functions in cells including embryonic development, maintenance of progenitor cell fate, cell cycle regulation, and potentially cancer development. The results of this study suggest that any potential dependence on MELK in cancer is not easily modeled using common cell culture conditions. Another important conclusion is the recognition that none of the tools in our study are perfectly specific, including the selective inhibitors and CRISPR-based technologies. Combined use of genetic, pharmacological and chemical genetic perturbations is required to reach conclusions regarding target dependence. For example, initial excitement generated by OTSSP167 and the effects of short hairpins on the proliferation of BBC cells were not recapitulated using selective MELK inhibitors and CRISPR-based technologies. By confirming that OTSSP167 and the MELK-targeting hairpins present off-target effects that cause growth defects, we deduced that MELK inhibition or depletion alone is not sufficient to impede the growth of BBC cell lines, or at a minimum, of MDA-MB-468 cells. Notably, a missense mutation (heterozygous G20V) in *MELK* in MDA-MB-468 line has been reported (*Forbes et al., 2017*), which may complicate data interpretation. Therefore, further studies need to be performed in other cancer cell lines. With the current data, we cannot rule out the possibility that MELK may play a crucial role for cancer maintenance in vivo, which would

only manifest in more context-dependent assays. Using clonogenic assays, an approach to measure the capability of single cells to grow into colonies, we recently demonstrate that both RNAi and CRISPR/Cas9-mediated gene editing impair the clonogenic growth of cancer cells, but cause minimal effects on regular cell growth (Wang et al., submitted to eLife). Our results also hinted at potential synthetic lethal relationships between MELK and other targets that are waiting to be explored. We expect that the novel chemical and chemical-genetic tools that we have generated will be useful in search of MELK-dependent biology.

## Materials and methods

### Tissue culture

MDA-MB-468 (RRID: CVCL_0419, ATCC) and MELK−/− MDA-MB-468 cells were authenticated using small tandem repeat (STR) profiling (DFCI Molecular Diagnostics Laboratory, Boston). The second source of MDA-MB-468 cells was a kind gift from Prof. Kun Ping Lu. All cell lines used in this study were routinely examined to be free of mycoplasma using the MycoAlert mycoplasma detection kit (Lonza, LT07-318). Cell lines HEK293T (RRID: CVCL_0063; ATCC) and MCF7 (RRID: CVCL_0031; ATCC) cells were maintained in DMEM (4.5 g/L D-glucose, L-glutamine, 110 mg/mL sodium pyruvate; Thermo Fisher Scientific) supplemented with 10% FBS (Thermo Fisher Scientific) and 100 U/mL penicillin-streptomycin (Thermo Fisher Scientific). Other cell lines including (RRID: CVCL_0419, ATCC), BT-549 (RRID: CVCL_1092; ATCC), HCC70 (RRID: CVCL_1270; ATCC), T-47D (RRID: CVCL_0553; ATCC) and ZR-75-1 (RRID: CVCL_0588; ATCC), were cultured in RPMI medium 1640 (L-glutamine; Thermo Fisher Scientific) supplemented with 10% FBS (Thermo Fisher Scientific) and 100 U/mL penicillin-streptomycin (Thermo Fisher Scientific). All cell lines were cultured at 37°C in a humidified chamber in the presence of 5% $CO_2$.

### Plasmids

Tet-pLKO-puro was a gift from Dmitri Wiederschain (Addgene plasmid # 21915). pHR-SFFV-KRAB-dCas9-P2A-mCherry (Addgene plasmid # 60954) and pU6-sgRNA EF1Alpha-puro-T2A-BFP (Addgene plasmid # 60955) were gifts from Jonathan Weissman. All oligonucleotides were ordered from IDT (Coralville, Iowa). To construct a Cas9/sgRNA expression system for MELK deletion, we annealed oligonucleotides targeting exon 4 or 5 of the *MELK* gene and directly ligated them with a linearized CRISPR nuclease vector (OFP reporter) from the GeneArt CRISPR nuclease vector kit (A21174, Life Technologies, Carlsbad, CA). For the expression of FKBP12$^{F36V}$-MELK(sg3R), we used a modified pLEX_305 (Addgene #41390) construct that contains a FKBP12$^{F36V}$-2xHA tag at the N-terminus (*Erb et al., 2017*). Human MELK or MELK(sg3R) in a pDONR223 vector was cloned into the modified pLEX_305 construct using Gateway cloning (Life Technologies). The mutation in MELK (sg3R) that confers resistance to sgMELK-3 was generated via Q5 site-directed mutagenesis kit (New England Biolabs). To generate doxycycline-inducible shRNA vectors targeting MELK, oligonucleotides selected from the RNAi Consortium shRNA Library or previously reported were annealed, and directly ligated with gel-purified tet-pLKO-puro backbone digested with AgeI and EcoRI. The H1 based doxycycline-inducible sgRNA vector (tet-pLKO-sgRNA-puro) that coexpresses the Tet repressor (TetR) from the hPGK promoter was modified from tet-pLKO-puro (Addgene plasmid # 21915) by replacing the sequences encoding for shRNA with sgRNA. Briefly, a pair of overlapping primers that encodes for an AjuI cloning site and the same sgRNA constant region as in pU6-sgRNA EF1Alpha-puro-T2A-BFP (Addgene plasmid # 60955) was amplified by PCR and digested with AgeI and EcoRI, followed by ligation with tet-pLKO-puro digested with AgeI and EcoRI. To construct doxycycline-inducible sgRNA plasmids that target MELK, oligonucleotides targeting the transcriptional start site of *MELK* were annealed and directly ligated with gel-purified tet-pLKO-sgRNA-puro backbone digested with AjuI. To complement the cut sites generated by AjuI, the oligonucleotides should contain overhangs as shown: 5'-CNNNN…..NNNNGTTTA-3' (forward) and 5'-NNNN…. NNNNGGTGTC-3' (reverse). The sequence NNNN….NNNN represents the guide sequence and one base 'G' needs to be added at the 5' end for efficient initiation of Pol III-mediated transcription if it does not already exist. All inserts were verified by Sanger sequencing (Genewiz), and the sequences of all oligonucleotides relevant for cloning are listed in *Supplementary file 2*. Plasmids were amplified in NEB 10-beta or NEB Stable competent E. coli.

## Lentiviral transduction

Lentiviruses were generated by transfecting 1.2 million HEK293T cells seeded one day prior in a T25 flask, with 2 μg lentiviral transfer plasmid, 0.5 μg packaging plasmid (pMD2.G), and 1.5 μg packaging plasmid (pCMV-dR8.91 or psPAX2). The plasmids were mixed in 400 μl of serum-free DMEM or Opti-MEM, and added with 12 μL of polyethyleneimine (PEI). After 15 minutes of incubation at room temperature, the DNA-PEI solution was added drop-wise to the HEK293T cells. Viral supernatant was collected 48 hours and 72 hours after transfection, filtered through 0.45 μm membrane, and added to target cells in the presence of 8 μg/ml polybrene (Millipore, Billerica, MA). Cells were selected with antibiotics starting 72 hours after initial infection. MDA-MB-468 cells were selected with 0.5 μg/mL of puromycin, and maintained under the same concentration for culture.

## Cell line construction

MDA-MB-468 cells that express KRAB-dCas9-P2A-mCherry were generated by lentiviral transduction and enriched by FACS sorting as a polyclonal population. MDA-MB-468 cells that express N-terminally tagged FKBP12$^{F36V}$-MELK(sg3R), doxycycline-inducible shRNA, or MDA-MB-468-KRAB-dCas9 cells that express constitutive or doxycycline-inducible sgRNA were generated by lentiviral transduction followed by puromycin selection (0.5 μg/mL). For generating MELK knockout cell clones, we transfected MDA-MB-468 or MDA-MB-468-FKBP12$^{F36V}$-MELK(sg3R) cells with a Cas9/sgMELK-3 plasmid using the Neon transfection system (1150V, 30 ms, 2 pulses) (Thermo Fisher Scientific). Cells expressing Cas9/sgMELK-3 were isolated by FACS for OFP +cells, and seeded sparsely in 10 cm plates. Single cell clones were selected and transferred to 96-well plates. Cell clones that survived the transfer were further expanded for genotyping and immunoblotting analysis to confirm the status of MELK.

## Genotyping

Genomic DNA was extracted from the cells transfected with Cas9/sgMELK-3 or from the expanded clonal cell lines using PureLink Genomic DNA Kits (Thermo Fisher Scientific). The genomic loci of interest were amplified by PCR using the primers listed in *Supplementary file 2*. Amplified PCR products were purified using QIAquick PCR Purification Kit, and sequenced using the same forward primer that was used for PCR amplification (Genewiz). The genetic status was determined with help of TIDE analysis (*Brinkman et al., 2014*) and direct interpretation of the ab1 files.

## Antibodies

Primary antibodies used in this study include anti-MELK (EPR3981, Abcam or GeneTex), anti-α-tubulin (RRID:AB_1904178; 3873S, Cell Signaling Technology), anti-GAPDH (RRID:AB_561053; 2118S, Cell Signaling Technology), and anti-Vinculin (RRID:AB_10604160; SAB4200080, Sigma). Secondary antibodies used were IRDye680-conjugated goat anti-rabbit IgG (RRID:AB_10956166; Cat# 926–68071, LI-COR Biosciences) and IRDye800-conjugated goat anti-mouse IgG (RRID:AB_621842; Cat# 926–32210, LI-COR Biosciences).

## Immunoblotting

Cells were lysed with RIPA buffer (50 mM Tris-HCl, 150 mM NaCl, 1% NP-40, 0.5% sodium deoxycholate, and 0.1% sodium dodecyl sulfate, pH 7.4) supplemented with EDTA-free protease inhibitor cocktails (cOmplete, Roche) and phosphatase inhibitor cocktails (PhosSTOP, Roche) at 4°C for at least 30 min. Clear lysates were collected after maximal speed centrifugation at 4°C for 10 minutes, and were analyzed for protein concentration using Pierce BCA protein assay (Thermo Fisher Scientific). Lysate concentrations are normalized, mixed with 4X NuPAGE LDS sample buffer (10% 2-mercaptoethanol added freshly; Thermo Fisher Scientific), and denatured at 95°C for 10 min. Equal amount of protein (20–40 μg) was resolved by Bolt 4–12% Bis-Tris Plus gels, and then transferred onto a nitrocellulose membrane. The membrane was blocked with 5% non-fat milk in TBS-T (TBS with 0.1% Tween-20), followed by incubation with primary antibodies at 4°C overnight. Next day, after washing with TBS-T, the membrane was incubated with fluorophore-conjugated secondary antibodies for 1 hour at room temperature. The membrane was then washed and scanned with an Odyssey Infrared scanner (Li-Cor Biosciences, Lincoln, NE). Note that in commercial gradient SDS-PAGE gels (majority of current study), MELK consistently appears as one band. While tested in 8%

homemade SDS polyacrylamide gel (EC-890, 30% ProtoGel, National Diagnostics), MELK presents as two sharp bands that are close to each other (e.g. *Figure 2A*).

## Cellular kinase engagement assay (Kinativ)

MDA-MB-468 cells on 10 cm plates were treated with HTH-01-091 at indicated concentrations and MG132 (10 μM) for an hour. After washing, each plate was lysed with 500 μL of Pierce IP lysis buffer supplemented with EDTA-free protease inhibitor cocktails (cOmplete, Roche) and phosphatase inhibitor cocktails (PhosSTOP, Roche). For each sample of 500 μL cleared lysate, 10 μL of 1 M $MgCl_2$ was added. Reconstitute ATP-desthiobiotin probe with water into 0.25 mM and add 10 μL to each sample. After 10 min, quench the labeling reaction by adding 500 μL of 8 M Urea/IP lysis buffer. To pulldown the labeled proteins, 50 μL of Pierce streptavidin agarose beads were used for each sample. The beads were boiled at 95°C for 10 min and analyzed by immunoblotting.

## Cell proliferation assay

For monitoring the growth rates of MDA-MB-468 cells (parental, a WT clone and a MELK−/− clone), 2000 cells/well were seeded in 96-well plates on Day 0. On Day 1, 3 and 5, the relative number of cells was assessed using CellTiter-Glo (Promega cat# G7571) as described in product manual by luminescence measurements on an Envision plate reader (PerkinElmer). Reagents were warmed up to room temperature and the incubation time was fixed to minimize variations from day to day. For testing the sensitivity of breast cancer cells to MELK inhibitors at different doses, cells were plated in 96-well plates at 2000 (3-day growth) or 1000 (7-day growth) cells/well in fresh media and treated with OTSSP167, MRT199665, HTH-01-091, MELK-T1, NVS-MELK8a or DMSO at the indicated concentrations the next day. Antiproliferative effects of compounds were assessed using CellTiter-Glo (Promega cat# G7571). $IC_{50}$ values were determined using GraphPad Prism 6 nonlinear regression curve fit. For assessing the effects of FKBP12$^{F36V}$-MELK(sg3R) degradation or MELK knockdown by doxycycline-inducible shRNA or sgRNA on proliferation, 10,000 cells/well were seeded in 6- or 12-well plates. On the next day, we added dTAG-47 (500 nM) or doxycycline (100 ng/mL) to the treated wells, which are refreshed every 3 or 2 days, respectively. After 9 days, cells were fixed with formaldehyde and stained with crystal violet (0.05% wt/vol). The plates were imaged using the Gel Doc XR + Documentation System (Biorad) and quantified using ColonyArea, an ImageJ plugin that quantifies cell staining areas and intensities (*Guzmán et al., 2014*).

## Inhibitor-induced MELK degradation

MDA-MB-468 cells ($0.25 \times 10^6$ cells/well) were seeded in 12-well plates the day before experiment. After treatment with MELK inhibitors, cells were harvested and analyzed by immunoblotting. For the rescue experiments, cells were pretreated with carfilzomib (0.4 μM) or MLN4924 (1 μM) for 4 hours before treatment with MELK inhibitors.

## Cell cycle analysis and mitotic arrest

For cell cycle analysis, cells were harvested, washed once in ice-cold phosphate buffered saline (PBS), and fixed overnight at –20°C with 80% ethanol in PBS. Cells were washed three times with PBS, and suspended in PBS containing 0.1% Triton X-100, 25 μg/mL propidium iodide (PI, Molecular Probes), and 0.2 mg/mL RNase A (Sigma). Samples were stained at 4°C overnight and stored at the same temperature until analysis by LSR Fortessa (BD Biosciences) flow cytometer. Results were analyzed using FlowJo (Treestar).

To arrest MDA-MB-468 cells in mitosis, cells underwent a single thymidine block (2 mM thymidine, 16–24 hours), a 3-hour release, followed by treatment with 10 μM S-trityl-L-cysteine for 10 hours. Roughly 50% of MDA-MB-468 cells would float and arrest in mitosis using this method.

## RT-qPCR analysis

Total RNA was extracted from cultured cells with RNeasy Plus Mini kit (Qiagen). 1 μg of the total RNA was reversely transcribed using iScript Reverse Transcription Supermix (Bio-Rad). cDNA were analyzed quantitatively using Power SYBR Green PCR Master Mix (Applied Biosystems) on an ABI7300 Real-time PCR system. Primers used were listed in *Supplementary file 2*. Cycling conditions were 95°C for 15 minutes, followed by 40 cycles of 15 seconds at 94°C, 30 seconds at 55°C

and 30 seconds at 72°C. Ct values were generated using the default analysis settings. $\Delta$Ct was defined as Ct $_{gene\ of\ interest}$ − Ct $_{actin}$. $\Delta\Delta$CT was defined as $\Delta$Ct $_{treated\ sample}$ − $\Delta$Ct $_{control\ sample}$. Relative quantification was calculated as $2^{-\Delta\Delta CT}$.

## Synthesis of HTH-01-091

**Chemical structure 1.** Ethyl 6-chloro-4-((*trans*-4-((dimethylamino)methyl)cyclohexyl)amino)quinoline-3-carboxylate.
DOI: https://doi.org/10.7554/eLife.26693.024

## Ethyl 6-chloro-4-((*trans*-4-((dimethylamino)methyl)cyclohexyl)amino)quinoline-3-carboxylate

In a stirring 1,4-dioxane solution (8 ml) of ethyl 4,6-dichloroquinoline-3-carboxylate (1 equiv., 0.716 mmol), *trans*-4-((dimethylamino)methyl)cyclohexan-1-amine diacetic acid (1 equiv., 0.716 mmol) and N,N-diisopropylethylamine (10 equiv., 7.16 mmol) were added and allowed to dissolve. The resulting solution was heated up to 90°C, and stirred for 12 hours before cooling to room temperature. The solvent was removed under reduced pressure, and the resultant crude was purified by Flash Column Chromatography on silica gel with 0–10% $CH_2Cl_2$/methanol (1.75N ammonia) gradient to give the desired product.

MS(ESI) calculated for $C_{21}H_{29}ClN_3O_2$ [M + H]$^+$, 390; found 390.

**Chemical structure 2.** Preparation of (6-chloro-4-((*trans*-4-((dimethylamino)methyl)cyclohexyl)amino)quinolin-3-yl)methanol.
DOI: https://doi.org/10.7554/eLife.26693.025

## Preparation of (6-chloro-4-((*trans*-4-((dimethylamino)methyl)cyclohexyl)amino)quinolin-3-yl)methanol

To a THF (9 mL) solution of ethyl 6-chloro-4-((*trans*-4-((dimethylamino)methyl)cyclohexyl)amino)quinoline-3-carboxylate (1 equiv., 0.308 mmol) purged with argon and sitting in an ice bath, lithium aluminium hydride (70 mg, 6 equiv., 1.85 mmol) was slowly added in three portions. The reaction mixture was allowed to warm up to room temperature and stirred overnight. The reaction was quenched by an addition of 70 µL of water, followed by 70 µL of 15% NaOH(aq) and 210 µL of water. Filter the crude through Celite. Purification was performed by Flash Column Chromatography on silica gel with 0–20% $CH_2Cl_2$/methanol (1.75N ammonia) gradient to give the desired compound.

MS(ESI) calculated for $C_{19}H_{27}ClN_3O$ [M + H]$^+$, 348; found 348.

**Chemical structure 3.** 6-chloro-3-(((2,4-dimethoxybenzyl)amino)methyl)-*N*-(*trans*-4-((dimethylamino)methyl)
cyclohexyl)quinolin-4-amine.
DOI: https://doi.org/10.7554/eLife.26693.026

## 6-chloro-3-(((2,4-dimethoxybenzyl)amino)methyl)-*N*-(*trans*-4-((dimethylamino) methyl)cyclohexyl)quinolin-4-amine

To a dichloromethane (2.5 mL) solution of (6-chloro-4-((*trans*-4-((dimethylamino)methyl)cyclohexyl) amino)quinolin-3-yl)methanol (89 mg, 1 equiv., 0.256 mmol) was added manganese oxide (445 mg, 5 mass equiv.) and stirred for 3 hours at room temperature. The reaction mixture was filtered and concentrated to afford the crude product, 6-chloro-4-((*trans*-4-((dimethylamino)methyl)cyclohexyl)amino) quinoline-3-carbaldehyde, which was used in the next step without further purification.

To a tetrahydrofuran (1 mL) solution containing 6-chloro-4-((*trans*-4-((dimethylamino)methyl)cyclo-hexyl)amino)quinoline-3-carbaldehyde (1 equiv., 0.05 mmol), (2,4-dimethoxyphenyl)methanamine (2.0 equiv., 0.10 mmol) and one drop of acetic acid was added sodium triacetoxyborohydride (5.0 equiv., 0.25 mmol). Stir at room temperature overnight. The reaction was worked up in water (10 mL) and dichloromethane (15 mL × 3). The organic layers were collected and washed with brine solution (10 mL). Purification was performed by Flash Column Chromatography on silica gel with 0–10% $CH_2Cl_2$/methanol (1.75N ammonia) gradient to afford the desired compound.

MS(ESI) calculated for $C_{28}H_{38}ClN_4O_2$ [M + H]$^+$, 497; found 497.

**Chemical structure 4.** Preparation of 9-chloro-3-(2,4-dimethoxybenzyl)−1-(*trans*-4-((dimethylamino)methyl)
cyclohexyl)−3,4-dihydropyrimido[5,4 *c*]quinolin-2(1*H*)-one.
DOI: https://doi.org/10.7554/eLife.26693.027

## Preparation of 9-chloro-3-(2,4-dimethoxybenzyl)−1-(*trans*-4-((dimethylamino) methyl)cyclohexyl)−3,4-dihydropyrimido[5,4 *c*]quinolin-2(1*H*)-one

To a dichloromethane solution containing 6-chloro-3-(((2,4-dimethoxybenzyl)amino)methyl)-*N*-(*trans*-4-((dimethylamino)methyl)cyclohexyl)quinolin-4-amine (1.0 equiv., 0.037 mmol) and diisopropylethyl-amine (3.0 equiv., 0.111 mmol) in ice bath was added triphosgene (1.0 equiv., 0.037 mmol). Stir for 1 hour. Remove the solvent under reduced pressure. Purification was performed by Flash Column Chromatography on silica gel with 0–10% $CH_2Cl_2$/methanol (1.75N ammonia) gradient to afford the desired compound.

MS(ESI) calculated for $C_{29}H_{36}ClN_4O_3$ [M + H]$^+$, 523; found 523.

**Chemical structure 5.** Preparation of 9-(3,5-dichloro-4-hydroxyphenyl)−1-(*trans*-4-((dimethylamino)methyl)
cyclohexyl)−3,4-dihydropyrimido[5,4 *c*]quinolin-2(1*H*)-one (HTH-01-091).
DOI: https://doi.org/10.7554/eLife.26693.028

## Preparation of 9-(3,5-dichloro-4-hydroxyphenyl)-1-(*trans*-4-((dimethylamino) methyl)cyclohexyl)-3,4-dihydropyrimido[5,4 *c*]quinolin-2(1*H*)-one (HTH-01-091)

In a 1,4-dioxane/sat. Na$_2$CO$_3$(aq)(3:1) solution (0.8 mL) containing 9-chloro-3-(2,4-dimethoxybenzyl)−1-(*trans*-4-((dimethylamino)methyl)cyclohexyl)−3,4-dihydropyrimido[5,4 *c*]quino-lin-2(1*H*)-one (1.0 equiv., 0.015 mmol) was added (3,5-dichloro-4-hydroxyphenyl)boronic acid (1.5 equiv., 0.023 mmol) and 2-di-*tert*-butylphosphino-2',4',6'-triisopropylbiphenyl (10% equiv., 0.0015 mmol). The reaction was purged thoroughly with argon, to which 10% equivalent of bis(tripheylphos-phine)palladium(II) dichloride (0.0015 mmol) was added. The reaction was heated to 85°C and con-tinue stirring for 2 hours. The reaction was worked up in water (10 mL) and CHCl$_3$:iPrOH(4:1) (15 mL ×3). The organic layers were collected and washed with brine solution (10 mL). The solvent was removed under reduced pressure to afford the crude product 9-(3,5-dichloro-4-hydroxyphenyl)−3-(2,4-dimethoxybenzyl)−1-(*trans*-4-((dimethylamino)methyl)cyclohexyl)−3,4-dihydropyrimido[5,4 *c*]qui-nolin-2(1*H*)-one, which was used in the next step without further purification.

MS(ESI) calculated for C$_{35}$H$_{39}$Cl$_2$N$_4$O$_4$ [M + H]$^+$, 649; found 649.

9-(3,5-Dichloro-4-hydroxyphenyl)−3-(2,4-dimethoxybenzyl)−1-(*trans*-4-((dimethylamino)methyl) cyclohexyl)−3,4-dihydropyrimido[5,4 *c*]quinolin-2(1*H*)-one (1.0 equiv., 0.015 mmol) was stirred in a trifluoroacetic acid/dichloromethane (1:1, 1 mL) solution overnight. The solvent was removed under reduced pressure, and the crude was purified by reverse-phase prep-HPLC (C18) using water (0.05% trifluoroacetic acid)/methanol (0.05% trifluoroacetic acid) gradient to afford the compound HTH-01-091 as a trifluoroacetic salt.

$^1$H NMR (400MHz, DMSO) δ 10.63–10.49 (br, 1H), 9.14–9.04 (br, 1H), 8.78 (s, 1H), 8.19 (dd, *J* = 8.6, 1.8 Hz, 1H), 8.10 (d, *J* = 8.6 Hz, 1H), 8.09 (d, *J* = 1.8 Hz, 1H), 7.84 (s, 2H), 7.63 (s, 1H), 4.40 (s, 2H), 3.85 (m, 1H), 2.89 (t, *J* = 6.3 Hz, 2H), 2.77 (s, 3H), 2.76 (s, 3H), 2.61 (m, 2H), 2.18 (m, 2H), 1.87 (m, 2H), 1.80 (m, 1H), 1.07 (m, 2H).

MS(ESI) calculated for C$_{26}$H$_{29}$Cl$_2$N$_4$O$_2$ [M + H]$^+$, 499; found 499.

## Synthesis of dTAG-36

**Chemical structure 6.** *tert*-butyl (8-((2-(2,6-dioxopiperidin-3-yl)−1,3-dioxoisoindolin-4-yl)amino)octyl)carbamate.
DOI: https://doi.org/10.7554/eLife.26693.029

## *tert*-butyl (8-((2-(2,6-dioxopiperidin-3-yl)−1,3-dioxoisoindolin-4-yl)amino)octyl)carbamate

2-(2,6-dioxopiperidin-3-yl)−4-fluoroisoindoline-1,3-dione (481.6 mg, 1.74 mmol, 1 eq) and *tert*-butyl (8-aminooctyl)carbamate (467.7 mg, 1.91 mmol, 1.1 eq) were dissolved in NMP (8.7 mL, 0.2M). DIPEA (606 µL, 3.48 mmol, 2 eq) was added and the mixture was heated to 90°C. After 15 hours, the mixture was diluted with ethyl acetate and washed with 10% citric acid (aq), saturated sodium bicarbonate, water and three times with brine. The organic layer was dried over sodium sulfate, filtered and concentrated under reduced pressure. Purification by column chromatography (ISCO, 12 g column, 0–5% MeOH/DCM, 25 min gradient) gave the desired product as a yellow oil (0.55 g, 1.099 mmol, 63%).

**1H NMR** (500 MHz, chloroform-*d*) δ 8.00 (s, 1H), 7.53–7.46 (m, 1H), 7.09 (d, *J* = 7.1 Hz, 1H), 6.88 (d, *J* = 8.5 Hz, 1H), 6.23 (s, 1H), 4.92 (dd, *J* = 12.1, 5.2 Hz, 1H), 4.51 (s, 1H), 3.26 (q, *J* = 6.6 Hz, 2H), 3.11 (d, *J* = 5.9 Hz, 2H), 2.90 (d, *J* = 15.8 Hz, 1H), 2.83–2.72 (m, 2H), 2.15–2.11 (m, 1H), 1.65 (q, *J* = 7.1 Hz, 2H), 1.38 (d, *J* = 59.1 Hz, 19H).

**LCMS** 501.42 (M + H).

**Chemical structure 7.** 4-((8-aminooctyl)amino)−2-(2,6-dioxopiperidin-3-yl)isoindoline-1,3-dione.
DOI: https://doi.org/10.7554/eLife.26693.030

## 4-((8-aminooctyl)amino)−2-(2,6-dioxopiperidin-3-yl)isoindoline-1,3-dione

*tert*-butyl (8-((2-(2,6-dioxopiperidin-3-yl)−1,3-dioxoisoindolin-4-yl)amino)octyl)carbamate (0.55 g, 1.099 mmol, 1 eq) was dissolved in TFA (11 mL) and heated to 50°C. After 40 min, the mixture was cooled to room temperature, diluted with MeOH/DCM and concentrated under reduced pressure. The crude material was triturated with diethyl ether and dried under vacuum to give a cream colored solid (522.97 mg, 1.016 mmol, 93%).

**1H NMR** (500 MHz, Methanol-*d₄*) δ 7.59–7.51 (m, 1H), 7.04 (dd, *J* = 7.9, 1.7 Hz, 2H), 5.06 (dd, *J* = 12.4, 5.5 Hz, 1H), 3.34 (d, *J* = 7.0 Hz, 2H), 2.95–2.81 (m, 3H), 2.79–2.66 (m, 2H), 2.15–2.08 (m, 1H), 1.67 (tt, *J* = 12.2, 7.2 Hz, 4H), 1.43 (d, *J* = 22.2 Hz, 8H).

**LCMS** 401.39 (M + H)

**Chemical structure 8.** (2*S*)-(1*R*)−3-(3,4-dimethoxyphenyl)−1-(2-(2-((8-((2-(2,6-dioxopiperidin-3-yl)−1,3-dioxoisoindolin-4-yl)amino)octyl)amino)−2-oxoethoxy)phenyl)propyl 1-((*S*)−2-(3,4,5-trimethoxyphenyl)butanoyl)piperidine-2-carboxylate (dTAG-36).
DOI: https://doi.org/10.7554/eLife.26693.031

(2S)-(1R)−3-(3,4-dimethoxyphenyl)−1-(2-(2-((8-((2-(2,6-dioxopiperidin-3-yl)−1,3-dioxoisoindolin-4-yl)amino)octyl)amino)−2-oxoethoxy)phenyl)propyl 1-((S)−2-(3,4,5-trimethoxyphenyl)butanoyl)piperidine-2-carboxylate (dTAG-36)

4-((8-aminooctyl)amino)−2-(2,6-dioxopiperidin-3-yl)isoindoline-1,3-dione trifluoroacetate salt (10.3 mg, 0.020 mmol, 1 eq) was added to 2-(2-((R)−3-(3,4-dimethoxyphenyl)−1-(((S)−1-((S)−2-(3,4,5-tri-methoxyphenyl)butanoyl)piperidine-2-carbonyl)oxy)propyl)phenoxy)acetic acid (13.9 mg, 0.020 mmol, 1 eq) as a 0.1 M solution in DMF (200 microliters) at room temperature. DIPEA (10.5 microliters, 0.060 mmol, 3 eq) and HATU (7.6 mg, 0.020 mmol, 1 eq) were then added. After 30 hours, the mixture was diluted with EtOAc, and washed with 10% citric acid (aq), brine, saturated sodium bicarbonate, water and brine. The organic layer was dried over sodium sulfate, filtered and condensed. Purification by column chromatography (ISCO, 4 g silica column, 0–10% MeOH/DCM, 25 min gradient) gave the desired product as a yellow solid (16.5 mg, 0.0153 mmol, 77%).

$^1$**H NMR** (500 MHz, Methanol-$d_4$) δ 7.54 (dd, $J$ = 8.5, 7.2 Hz, 1H), 7.23 (td, $J$ = 8.1, 1.6 Hz, 1H), 7.06–6.99 (m, 2H), 6.88 (q, $J$ = 7.9, 7.3 Hz, 2H), 6.85–6.79 (m, 1H), 6.79–6.75 (m, 1H), 6.75–6.71 (m, 1H), 6.66–6.59 (m, 3H), 6.12 (dd, $J$ = 8.1, 6.0 Hz, 1H), 5.41 (d, $J$ = 4.3 Hz, 1H), 5.05 (dd, $J$ = 12.6, 5.5 Hz, 1H), 4.60–4.39 (m, 3H), 4.11 (d, $J$ = 13.4 Hz, 1H), 3.89–3.83 (m, 1H), 3.82–3.63 (m, 17H), 3.28 (d, $J$ = 6.9 Hz, 2H), 3.15 (tq, $J$ = 13.2, 6.7 Hz, 2H), 2.89–2.82 (m, 1H), 2.72 (ddd, $J$ = 14.4, 10.1, 3.7 Hz, 2H), 2.65–2.56 (m, 1H), 2.56–2.42 (m, 2H), 2.23 (t, $J$ = 12.1 Hz, 1H), 2.13–2.06 (m, 1H), 2.03 (ddd, $J$ = 13.7, 9.9, 5.0 Hz, 2H), 1.93 (tt, $J$ = 13.8, 6.4 Hz, 1H), 1.77–1.33 (m, 12H), 1.24–1.16 (m, 4H), 0.87 (t, $J$ = 7.4 Hz, 3H).

$^{13}$**C NMR** (126 MHz, MeOD) δ 174.76, 174.65, 172.52, 171.60, 170.84, 170.48, 169.30, 155.39, 154.62, 150.37, 148.83, 148.29, 138.04, 137.25, 136.89, 134.92, 133.92, 130.53, 129.87, 128.39, 123.14, 121.80, 117.97, 113.57, 113.05, 112.72, 111.73, 110.98, 106.66, 106.57, 105.96, 70.69, 68.05, 61.06, 56.58, 56.51, 56.45, 53.41, 51.00, 50.20, 45.01, 43.34, 40.07, 37.45, 32.22, 32.18, 30.25, 30.16, 29.33, 27.87, 27.75, 27.57, 26.39, 23.82, 21.96, 12.58.

**LCMS**: 1077.32 (M + H)

## Synthesis of dTAG-47

**Chemical structure 9.** *tert*-butyl (8-((2-(2,6-dioxopiperidin-3-yl)−1,3-dioxoisoindolin-5-yl)amino)octyl)carbamate.
DOI: https://doi.org/10.7554/eLife.26693.032

### *tert*-butyl (8-((2-(2,6-dioxopiperidin-3-yl)−1,3-dioxoisoindolin-5-yl)amino)octyl)carbamate

2-(2,6-dioxopiperidin-3-yl)−5-fluoroisoindoline-1,3-dione (294 mg, 1.06 mmol, 1 eq) and *tert*-butyl (8-aminooctyl)carbamate (286 mg, 1.17 mmol, 1.1 eq) were dissolved in NMP (5.3 mL, 0.2M). DIPEA (369 μL, 2.12 mmol, 2 eq) was added and the mixture was heated to 90℃. After 19 hours, the mixture was diluted with ethyl acetate and washed with water and three times with brine. The organic layer was dried over sodium sulfate, filtered and concentrated under reduced pressure. Purification by column chromatography (ISCO, 12 g column, 0–10% MeOH/DCM, 30 min gradient) gave the desired product as a brown solid (0.28 g, 0.668 mmol, 63%).

$^1$**H NMR** (500 MHz, Chloroform-$d$) δ 8.12 (s, 1H), 7.62 (d, $J$ = 8.3 Hz, 1H), 7.02 (s, 1H), 6.81 (d, $J$ = 7.2 Hz, 1H), 4.93 (dd, $J$ = 12.3, 5.3 Hz, 1H), 4.51 (s, 1H), 3.21 (t, $J$ = 7.2 Hz, 2H), 3.09 (d, $J$ = 6.4 Hz, 2H), 2.90 (dd, $J$ = 18.3, 15.3 Hz, 1H), 2.82–2.68 (m, 2H), 2.16–2.08 (m, 1H), 1.66 (p, $J$ = 7.2 Hz, 2H), 1.37 (d, $J$ = 62.3 Hz, 20H).

**LCMS** 501.41 (M + H)

**Chemical structure 10.** 5-((8-aminooctyl)amino)−2-(2,6-dioxopiperidin-3-yl)isoindoline-1,3-dione trifluoroacetate.
DOI: https://doi.org/10.7554/eLife.26693.033

## 5-((8-aminooctyl)amino)−2-(2,6-dioxopiperidin-3-yl)isoindoline-1,3-dione trifluoroacetate

*tert*-butyl (8-((2-(2,6-dioxopiperidin-3-yl)−1,3-dioxoisoindolin-5-yl)amino)octyl)carbamate (334.5 g, 0.668 mmol, 1 eq) was dissolved in TFA (6.7 mL) and heated to 50°C. After 1 hour, the mixture was cooled to room temperature, diluted with DCM and concentrated under reduced pressure. The crude material was triturated with diethyl ether and dried under vacuum to give a dark yellow foam (253.1 mg, 0.492 mmol, 74%).

[1]H NMR (500 MHz, Methanol-$d_4$) δ 7.56 (d, *J* = 8.4 Hz, 1H), 6.97 (d, *J* = 2.1 Hz, 1H), 6.83 (dd, *J* = 8.4, 2.2 Hz, 1H), 5.04 (dd, *J* = 12.6, 5.5 Hz, 1H), 3.22 (t, *J* = 7.1 Hz, 2H), 2.94–2.88 (m, 2H), 2.85–2.68 (m, 3H), 2.09 (ddd, *J* = 10.4, 5.4, 3.0 Hz, 1H), 1.70–1.61 (m, 4H), 1.43 (d, *J* = 19.0 Hz, 8H).

LCMS 401.36 (M + H)

**Chemical structure 11.** (2S)-(1R)−3-(3,4-dimethoxyphenyl)−1-(2-(2-((8-((2-(2,6-dioxopiperidin-3-yl)−1,3-dioxoisoindolin-5-yl)amino)octyl)amino)−2-oxoethoxy)phenyl)propyl 1-((S)−2-(3,4,5-trimethoxyphenyl)butanoyl)piperidine-2-carboxylate (dTAG-47).
DOI: https://doi.org/10.7554/eLife.26693.034

## (2S)-(1R)−3-(3,4-dimethoxyphenyl)−1-(2-(2-((8-((2-(2,6-dioxopiperidin-3-yl)−1,3-dioxoisoindolin-5-yl)amino)octyl)amino)−2-oxoethoxy)phenyl)propyl 1-((S)−2-(3,4,5-trimethoxyphenyl)butanoyl)piperidine-2-carboxylate (dTAG-47)

5-((8-aminooctyl)amino)−2-(2,6-dioxopiperidin-3-yl)isoindoline-1,3-dione trifluoroacetate salt (10.3 mg, 0.020 mmol, 1 eq) was added to 2-(2-((R)−3-(3,4-dimethoxyphenyl)−1-(((S)−1-((S)−2-(3,4,5-tri-methoxyphenyl)butanoyl)piperidine-2-carbonyl)oxy)propyl)phenoxy)acetic acid (13.9 mg, 0.020 mmol, 1 eq) as a 0.1 M solution in DMF (200 microliters) at room temperature. DIPEA (10.5 microliters, 0.060 mmol, 3 eq) and HATU (7.6 mg, 0.020 mmol, 1 eq) were then added. After 29.5 hours, the mixture was diluted with EtOAc, and washed with 10% citric acid (aq), brine, saturated sodium bicarbonate, water and brine. The organic layer was dried over sodium sulfate, filtered and condensed. Purification by column chromatography (ISCO, 4 g silica column, 0–10% MeOH/DCM, 25 min gradient) gave the desired product as a yellow solid (14.1 mg, 0.0131 mmol, 65%).

[1]H NMR (500 MHz, Methanol-$d_4$) δ 7.55 (d, *J* = 8.4 Hz, 1H), 7.26–7.20 (m, 1H), 6.99–6.93 (m, 1H), 6.89 (t, *J* = 7.7 Hz, 2H), 6.82 (dd, *J* = 8.4, 2.3 Hz, 2H), 6.77 (d, *J* = 7.5 Hz, 1H), 6.74 (d, *J* = 1.9 Hz, 1H), 6.63 (d, *J* = 9.6 Hz, 2H), 6.12 (dd, *J* = 8.1, 6.0 Hz, 1H), 5.40 (d, *J* = 4.3 Hz, 1H), 5.03 (dd, *J* = 13.1, 5.5 Hz, 1H), 4.57 (d, *J* = 14.9 Hz, 1H), 4.46–4.39 (m, 1H), 4.11 (d, *J* = 13.6 Hz, 1H), 3.86 (t, *J* = 7.3 Hz, 1H), 3.80–3.76 (m, 7H), 3.71–3.65 (m, 8H), 3.14 (ddt, *J* = 17.2, 13.3, 7.1 Hz, 4H), 2.90–

2.80 (m, 1H), 2.77–2.40 (m, 6H), 2.24 (d, $J$ = 13.8 Hz, 1H), 2.12–1.97 (m, 3H), 1.92 (dq, $J$ = 14.0, 7.8 Hz, 1H), 1.67 (ddt, $J$ = 54.1, 14.7, 7.1 Hz, 5H), 1.50 (dd, $J$ = 46.1, 14.1 Hz, 3H), 1.38 (dt, $J$ = 14.5, 7.1 Hz, 4H), 1.28–1.17 (m, 6H), 0.87 (t, $J$ = 7.3 Hz, 3H).

[13]C NMR (126 MHz, MeOD) δ 174.78, 174.69, 172.53, 171.71, 170.50, 169.66, 169.31, 156.22, 155.41, 154.62, 150.36, 148.83, 138.05, 136.90, 136.00, 134.93, 130.54, 128.40, 126.21, 123.14, 121.82, 117.94, 116.62, 113.58, 113.05, 112.73, 106.59, 70.69, 68.05, 61.06, 56.59, 56.51, 56.45, 53.42, 50.99, 50.31, 45.01, 44.09, 40.07, 37.44, 32.22, 32.17, 30.38, 30.32, 30.18, 29.84, 29.32, 28.05, 27.80, 27.58, 26.38, 23.87, 21.95, 12.57.

LCMS: 1077.35 (M + H)

## Crystal structure determination

To obtain the structure of MELK in complex with MRT199665, the MELK kinase domain (residues 1–340) was expressed as a His6-GST-fusion using a baculovirus/insect cell expression system. After purification and cleavage of the affinity tags, purified MELK protein was concentrated to 8 mg/ml in buffer containing 20 mM Tris (pH 8.0), 300 mM NaCl, 5% glycerol and 4 mM DTT. Well-ordered apo MELK crystals were obtained in hanging drops over a reservoir containing 100 mM Bis-Tris (pH 7.0), 10% (w/v) PEG3350, 200 mM NaCl, 10% glycerol and 10 mM DTT. The complex structure was made by soaking apo MELK crystals for 84 hours in 100 μM MRT199665 in cryo-buffer (100 mM Bis-Tris (pH 7.0), 10% (w/v) PEG3350, 200 mM NaCl, 25% glycerol, 10 mM DTT). Diffraction data were collected at the Argonne National Laboratory ID24E beamline. The structure was solved by molecular replacement with PHENIX (*Adams et al., 2010*) using the unliganded structure as a search model. Coordinates for MRT199665 were generated using PRODRG (*Schüttelkopf and van Aalten, 2004*), and were fit using PHENIX. Further refitting and refinement of the model were performed with Coot (*Emsley and Cowtan, 2004*) and PHENIX.

To obtain the structure of MELK in complex with HTH-01-091, a construct of human MELK covering residues 2–333 in the pTRXHGST vector was overexpressed in E. coli BL21 (DE3) in LB medium in the presence of 50 mg/ml of kanamycin. After purification and cleavage of the affinity tags, purified MELK protein was concentrated to 10 mg/ml in 20 mM HEPES (pH 7.5), 200 mM NaCl, 5% glycerol, and 5 mM DTT. Two equivalences of HTH-01-091 (from a 10 mM in DMSO stock) was mixed with 250 μM protein and crystallized by sitting-drop vapor diffusion at 20°C in the following crystallization buffer: 25% (w/v) PEG3350, 0.2 M MgCl$_2$, 0.1 M Bis-Tris (pH 5.5). Crystals were transferred briefly into crystallization buffer containing 25% glycerol prior to flash-freezing in liquid nitrogen. Diffraction data from complex crystals were collected at Argonne National Laboratory ID24E beamline. Data sets were integrated and scaled using XDS (*Kabsch, 2010*). Structures were solved by molecular replacement using the program Phaser (*McCoy et al., 2007*) and the search model PDB entry 4BL1. The ligand was positioned and preliminarily refined using Buster and Rhofit (*Smart et al., 2012*). Iterative manual model building and refinement using PHENIX and COOT led to a model with excellent statistics (*Supplementary file 1*).

## Accession codes

PDB codes 5TWL and 5TX3 represent MELK crystal structure in complex with HTH-01-091 and MRT199665, respectively. The authors will release the atomic coordinates and experimental data upon article publication.

## Sample preparation for quantitative mass spectrometry analysis

MELK−/− MDA-MB-468-FKBP12$^{F36V}$-MELK cells were treated in duplicates with DMSO for 30 minutes or 60 minutes, or in triplicates with dTAG-7 (250 nM) for 30 minutes or 60 minutes. Cells were washed with ice-cold PBS once, collected by scraping and centrifugation, and lysed in 1 mL of lysis buffer (8 M urea, 1% SDS, 50 mM Tris pH8.5 supplemented with protease and phosphatase inhibitors). A micro-BCA assay (Pierce) was used to determine the protein concentration of each sample. A total of 1200 μg of proteins from each sample was depleted of abundant proteins using two Pierce Top 12 protein depletion spin columns. Eluted proteins were combined, and protein concentration was determined by micro-BCA. Proteins were precipitated on ice for 1 hour with 13% trichloroacetic acid (TCA), washed twice with acetone, and pellets were allowed to air dry. Proteins were resuspended in 6 M urea, 50 mM Tris pH 8.5, reduced with 5 mM dithiothreitol (DTT) at room

temperature for 1 hour, and alkylated with 15 mM iodoacetamide in the dark at room temperature for 1 hour. Alkylation was quenched with DTT. Urea concentration was reduced to 4 M and proteins were digested with LysC (1:50; enzyme:protein) for 6 hours at 25°C. The LysC digestion was diluted to 1 M urea, 50 mM Tris pH 8.5 and then digested with trypsin (1:100; enzyme:protein) overnight at 37°C. Peptides were desalted using a $C_{18}$ solid phase extraction cartridges as previously described (*Weekes et al., 2014*) and dried by SpeedVac. Dried peptides were resuspended in 200 mM EPPS pH 8.0. Peptide quantification was performed using the micro-BCA assay (Pierce). The same amount of peptide from each condition was labeled with tandem mass tag (TMT) reagent (1:4; peptide:TMT label) (Pierce). The 10-plex labeling reactions were performed for 2 hour at 25°C. Modification of tyrosine residue with TMT was reversed by the addition of 5% hydroxyl amine for 15 min at 25°C. The reaction was quenched with 0.5% trifluoroactic acid (TFA) and samples were combined at a 1:1:1:1:1:1:1:1:1:1 ratio. Combined samples were desalted and offline fractionated into 24 fractions as previously described (*McAlister et al., 2014*).

## Liquid chromatography-MS3 spectrometry (LC-MS/MS)

12 of the 24 peptide fractions from the basic reverse phase fractionation (every other fraction) were analyzed with an LC-MS3 data collection strategy (*McAlister et al., 2014*) on an Orbitrap Fusion mass spectrometer (Thermo Fisher Scientific) equipped with a Proxeon Easy nLC 1000 for online sample handling and peptide separations. Approximately, 5 μg of peptide resuspended in 5% formic acid +5% acetonitrile was loaded onto a 100 μm inner diameter fused-silica micro capillary with a needle tip pulled to an internal diameter less than 5 μm. The column was packed in-house to a length of 35 cm with a $C_{18}$ reverse phase resin (GP118 resin 1.8 μm, 120 Å, Sepax Technologies). The peptides were separated using a 180 min linear gradient from 3% to 25% buffer B (100% acetonitrile (ACN) +0.125% formic acid) equilibrated with buffer A (3% ACN +0.125% formic acid) at a flow rate of 400 nL/min across the column. The scan sequence for the Fusion Orbitrap began with an MS1 spectrum (Orbitrap analysis, resolution 120,000, 400–1400 m/z scan range, AGC target $2 \times 10^5$, maximum injection time 100 ms, dynamic exclusion of 90 s). The 'Top10' precursors was selected for MS2 analysis, which consisted of CID (quadrupole isolation set at 0.5 Da and ion trap analysis, AGC $8 \times 10^3$, NCE 35, maximum injection time 150 ms). The top ten precursors from each MS2 scan were selected for MS3 analysis (synchronous precursor selection), in which precursors were fragmented by HCD prior to Orbitrap analysis (NCE 55, max AGC $1 \times 10^5$, maximum injection time 150 ms, resolution 60,000.

## LC-MS3 data analysis

A suite of in-house software tools were used to for. RAW file processing and controlling peptide and protein level false discovery rates, assembling proteins from peptides, and protein quantification from peptides as previously described. MS/MS spectra were searched against a Uniprot human database (February 2014) with both the forward and reverse sequences. Database search criteria are as follows: tryptic with two missed cleavages, a precursor mass tolerance of 50 ppm, fragment ion mass tolerance of 1.0 Da, static alkylation of cysteine (57.02146 Da), static TMT labeling of lysine residues and N-termini of peptides (229.162932 Da), and variable oxidation of methionine (15.99491 Da). TMT reporter ion intensities were measured using a 0.003 Da window around the theoretical m/z for each reporter ion in the MS3 scan. Peptide spectral matches with poor quality MS3 spectra were excluded from quantitation (<200 summed signal-to-noise across 10 channels and <0.5 precursor isolation specificity).

## Differential protein abundance analysis

The entire set of data were disclosed in *Figure 4—source data 1*, but we only presented the results from the 60-min time point in this report (*Figure 4—source data 2*), as the 30-min time point provided a similar conclusion with milder FKBP12$^{F36V}$-MELK degradation. All data analysis was carried out using the R statistical framework (*Core Team, 2016*). Reporter ion intensities were normalized (normalization factors calculated by dividing the summed intensities of each channel by the maximum value of all channels, and subsequently applied to each reporter ion value) and scaled (scaling the sum of reporter ion intensities to 100 for each protein/peptide) using in-house scripts. Proteins quantified with a minimum of 2 unique peptides were considered for downstream analysis. Log$_2$

transformed, scaled and normalized reporter ion intensities were analyzed using a linear model approach implemented in the *limma* package (RRID:SCR_010943), which is free and available online under Bioconductor (https://www.bioconductor.org/help/search/index.html?q=limma/) (*Ritchie et al., 2015*). Resulting data were subjected to a moderated t-test to assess statistical significance also implemented in the *limma* package. In the moderated t-test, the standard errors are more robust calculated using an empirical Bayes method inferring information across all proteins. The 'topTable' output from *limma* is provided as *Figure 4—source data 2*, and contains the values *LogFC* (log$_2$ transformed fold change of dTAG-7 to DMSO), *AveExpr* (not interpretable in the study design), *t* (moderated t-statistic), *P Value* (associated *P* Value), and *Adjusted P Value* (the *P* Value adjusted for multiple hypothesis testing). Data in *Figure 4—source data 2* were used for plotting *Figure 4D*.

## Statistical analysis

Statistical tests and the associated error bars are identified in the corresponding figure legends. Typical replicate numbers describe the number of technical replicates analyzed in a single experiment. 'Independent' replicate numbers describe the number of biological replicates, which were experiments performed on different days. Data met the assumptions for all tests used. Sample sizes were not predetermined using any statistical analyses.

## Acknowledgements

We thank Ryan C Kunz and Rachel B Rodrigues for performing the quantitative proteomics studies at the Thermo Fisher Scientific Center for Multiplexed Proteomics at Harvard Medical School. Eric S Fischer kindly provided the script and guidance for the analysis of multiplexed proteomics data. We also thank Eric S Wang and Nicholas Kwiatkowski for critical reading of the manuscript. We thank Dario R Alessi for facilitating our collaboration with the MRC Protein Phosphorylation and Ubiquitylation Unit and providing MRT199665. We thank Stephen Deangelo for growing and harvesting bacterial cultures. X-ray crystallography was conducted at the Advanced Photon Source on the Northeastern Collaborative Access Team beamlines (NIGMS P41 GM103403). This research was supported by Dana-Farber Cancer Institute's Accelerator Fund. DLB was a Merck Fellow of the Damon Runyon Cancer Research Foundation (DRG-2196-14). BN was supported by an American Cancer Society Postdoctoral Fellowship (PF-17-010-01-CDD).

## Additional information

### Competing interests

Hai-Tsang Huang, Tinghu Zhang, Yubao Wang, Jean J Zhao: An inventor on patent PCT Int. Appl. (2016), WO 2016141296 A1 20160909. covering the use of HTH-01-091. James E Bradner: A scientific founder of Syros Pharmaceuticals, SHAPE Pharmaceuticals, Acetylon Pharmaceuticals, Tensha Therapeutics (now Roche) and C4 Therapeutics and is the inventor on IP licensed to these entities. J. E.B. is now an executive and shareholder in Novartis AG. Nathanael S Gray: An inventor on patent PCT Int. Appl. (2016), WO 2016141296 A1 20160909. covering the use of HTH-01-091. A founder of C4 Therapeutics, which has licensed degrader related intellectual property from DFCI. The other authors declare that no competing interests exist.

### Funding

| Funder | Grant reference number | Author |
| --- | --- | --- |
| Dana-Farber Cancer Institute | Accelerator Fund | Hai-Tsang Huang<br>Tinghu Zhang<br>Yubao Wang<br>Jean J Zhao<br>Nathanael S Gray |
| Damon Runyon Cancer Research Foundation | DRG-2196-14 | Dennis L Buckley |

| American Cancer Society | PF-17-010-01-CDD | Behnam Nabet |

The funders had no role in study design, data collection and interpretation, or the decision to submit the work for publication.

## Author contributions

Hai-Tsang Huang, Conceptualization, Formal analysis, Investigation, Writing—original draft, Writing—review and editing; Hyuk-Soo Seo, Qing Li, Formal analysis, Investigation; Tinghu Zhang, Conceptualization, Supervision, Investigation; Yubao Wang, Conceptualization, Resources, Investigation; Baishan Jiang, Resources; Dennis L Buckley, Justin M Roberts, Shiva Dastjerdi, Georg E Winter, Resources, Methodology; Behnam Nabet, Resources, Methodology, Writing—review and editing; Joshiawa Paulk, James E Bradner, Methodology; Hilary McLauchlan, Jennifer Moran, Investigation; Michael J Eck, Supervision, Funding acquisition; Sirano Dhe-Paganon, Formal analysis, Supervision, Investigation; Jean J Zhao, Conceptualization, Supervision, Funding acquisition; Nathanael S Gray, Conceptualization, Supervision, Funding acquisition, Writing—review and editing

## Author ORCIDs

Hai-Tsang Huang (iD) http://orcid.org/0000-0002-4244-2304
Yubao Wang (iD) http://orcid.org/0000-0003-2703-3104
Justin M Roberts (iD) http://orcid.org/0000-0001-6112-7476
Nathanael S Gray (iD) http://orcid.org/0000-0001-5354-7403

## Decision letter and Author response

Decision letter https://doi.org/10.7554/eLife.26693.042
Author response https://doi.org/10.7554/eLife.26693.043

# Additional files

## Supplementary files

• Supplementary file 1. Crystallization conditions and data collection and refinement statistics for crystal structures.
DOI: https://doi.org/10.7554/eLife.26693.035

• Supplementary file 2. Oligonucleotides and primers. The oligos and primers necessary for all cloning, genotyping and RT-qPCR work in this study are listed.
DOI: https://doi.org/10.7554/eLife.26693.036

• Transparent reporting form
DOI: https://doi.org/10.7554/eLife.26693.037

## Major datasets

The following datasets were generated:

| Author(s) | Year | Dataset title | Dataset URL | Database, license, and accessibility information |
| --- | --- | --- | --- | --- |
| Qing Li, Hyuk-Soo Seo, Hai-Tsang Huang, Nathanael S Gray, Sirano Dhe-Paganon, Michael J Eck | 2017 | Structure of Maternal Embryonic Leucine Zipper Kinase | https://www.rcsb.org/pdb/explore/explore.do?structureId=5TX3 | Publicly available at the RCSB Protein Data Bank (accession no. 5TX3) |
| Hyuk-Soo Seo, Hai-Tsang Huang, Nathanael S Gray, Sirano Dhe-Paganon | 2017 | Structure of Maternal Embryonic Leucine Zipper Kinase | https://www.rcsb.org/pdb/explore/explore.do?structureId=5TWL | Publicly available at the RCSB Protein Data Bank (accession no. 5TWL) |

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
