## [Decision Letter]

Thank you for submitting your article "Chemical and Genetic Investigation of Targeting Maternal Embryonic Leucine Zipper Kinase in Basal-like Breast Cancer" for consideration by *eLife*. Your article has been favorably evaluated by Tony Hunter (Senior Editor) and two reviewers, one of whom, Jeffrey Settleman (Reviewer #1), is a member of our Board of Reviewing Editors. The following individual involved in review of your submission has agreed to reveal their identity: Benjamin F Cravatt (Reviewer #2).

The reviewers have discussed the reviews with one another and the Reviewing Editor has drafted this decision to help you prepare a revised submission.

Huang and co-authors have used several orthogonal approaches to evaluate the role of the MELK kinase in basal breast cancers. Their work extends previously reported findings by several groups indicating that MELK levels are increased in various cancers and that MELK seems to be required in several tested cancer cell lines based on RNAi as well as pharmacologic studies using a small molecule MELK inhibitor that is currently undergoing clinical investigation as an anti-cancer agent.

The approaches include the discovery and testing of a novel, more selective MELK kinase inhibitor, the use of CRISPR-based targeted gene knockouts of MELK, as well as a so-called "dTAG" method that enabled the acute and selective degradation of ectopically expressed MELK protein via pharmacologic treatment. The manuscript should be of broad interest to readership of *eLife*.

Overall, this is an exceptionally well executed and well documented study, which highlights the critical caveats associated with oncology target validation and provides an excellent example of the proper approach to validation of novel candidates. The findings convincingly demonstrate with strong likelihood that MELK is not required for the proliferation of the tested cancer cells and that the MELK inhibitor undergoing clinical evaluation is almost certainly not driving any efficacy via effects on MELK. However, a few important points should be addressed prior to publication:

1) The authors make the argument that the observed similar profile of weak (IC50 > 1 μM) anti-proliferative activity of their new MELK inhibitor in isogenic breast cancer cells in which MELK is either WT or CRISPR-deleted (Figure 2) supports the conclusion that specific pharmacologic inhibition of MELK does not affect these cells. However, the data shown in Figure 2 regarding target engagement make it difficult to determine whether, at concentrations of the inhibitor that "cover" the target, there is sufficient MELK selectivity to support that conclusion. It seems that, between 1 and 10 μM of the inhibitor, there are cell viability effects--which are presumably off-target effects. But this is also the concentration range where MELK seems to be engaged by the inhibitor (Figure 2). It would be helpful to include an additional more relevant pharmacodynamic readout of MELK suppression; for example, a change in a phospho-substrate, in order to generate a more robust cellular IC50 for target suppression and to enable a more compelling conclusion about the absence of any detectable effects of MELK inhibition on cell viability. If this is not possible, the authors should at least discuss this caveat regarding the conclusion of this part of the analysis. This discussion should also address what appears to be a substantial "cell shift" in potency of the inhibitor (at least 100-fold).

2) The authors have not demonstrated that the dTAG-MELK fusion retains biological function. While it probably does, in the absence of a formal demonstration of that, it is not possible to use these findings to support a definitive conclusion about the putative lack of adaptive MELK independency based on the negative data that is presented.

3) While it is great to see proteomic data confirming selective reductions of MELK in the dTAG degrader experiments, the actual magnitude of MELK reduction is very modest (~40%), which contrasts with the apparent complete loss of MELK expression observed by Western blotting (compare Figure 4). Presumably, this reflects that the proteomic data were acquired after only 1 hour of dTAG probe treatment. But, why was such an early time point selected for proteomic analysis of protein expression? It would have been arguably more interesting to see the proteomic profiles at 12 or 24 h post-dTAG probe treatment.

4) In the bar graph in Figure 5, the individual bars should be labeled to show which shRNA is associated with each bar (as was done in panel 5B).

5) The authors should cite a very recently published report in *eLife* (Lin et al., March 2017) that reaches similar conclusions regarding MELK's role in cancer cells.

6) The title should be revised to something that is better matched to the more definitive conclusions that are described in the Abstract.

---

## [Author Response]

[…] Overall, this is an exceptionally well executed and well documented study, which highlights the critical caveats associated with oncology target validation and provides an excellent example of the proper approach to validation of novel candidates. The findings convincingly demonstrate with strong likelihood that MELK is not required for the proliferation of the tested cancer cells and that the MELK inhibitor undergoing clinical evaluation is almost certainly not driving any efficacy via effects on MELK. However, a few important points should be addressed prior to publication:1) The authors make the argument that the observed similar profile of weak (IC50 > 1 μM) anti-proliferative activity of their new MELK inhibitor in isogenic breast cancer cells in which MELK is either WT or CRISPR-deleted (Figure 2) supports the conclusion that specific pharmacologic inhibition of MELK does not affect these cells. However, the data shown in Figure 2 regarding target engagement make it difficult to determine whether, at concentrations of the inhibitor that "cover" the target, there is sufficient MELK selectivity to support that conclusion. It seems that, between 1 and 10 μM of the inhibitor, there are cell viability effects--which are presumably off-target effects. But this is also the concentration range where MELK seems to be engaged by the inhibitor (Figure 2). It would be helpful to include an additional more relevant pharmacodynamic readout of MELK suppression; for example, a change in a phospho-substrate, in order to generate a more robust cellular IC50 for target suppression and to enable a more compelling conclusion about the absence of any detectable effects of MELK inhibition on cell viability. If this is not possible, the authors should at least discuss this caveat regarding the conclusion of this part of the analysis. This discussion should also address what appears to be a substantial "cell shift" in potency of the inhibitor (at least 100-fold).

We agree that the proliferation studies with inhibitors of varying degrees of selectivity for MELK do not conclusively demonstrate that inhibition of MELK does not contribute in part to their pharmacological effects. We have therefore removed the statements within the manuscript suggesting we have shown that specific pharmacologic inhibition of MELK does not affect the growth of these cells.

We have demonstrated that the inhibitors inhibit the proliferation of WT versus MELK-/- cell lines with similar potency, from which we conclude that MELK inhibition is not necessary to observe growth inhibition in the MELK-/- cells. Clearly this does not prove that inhibition of MELK is not important in WT cells and that the dependence on MELK is lost during the process of generating the MELK-/- cell lines. The best experiment to address whether there is a MELK-dependent pharmacological effect in the WT cells would be to perform a ‘rescue’ experiment by introducing a functional but inhibitor-resistant version of MELK. We used molecular modeling and site-directed mutagenesis to generate a number of MELK mutants but unfortunately were unsuccessful in identifying a mutation that retained WT kinase activity and conferred greater that 50-fold resistance to OTSSP167 and HTH-01-091. We have now included a discussion of this point in the Discussion section.

The second problem, as pointed out by the reviewers, is that we have not demonstrated selectivity on MELK over other targets inside cells. Indeed, the potency of HTH-01-091 evaluated by the Kinativ target engagement assay fell within the range where the cell viability starts to get affected by the potential off-targets of HTH-01-091. Kinativ target engagement assays may underestimate the potency of reversible inhibitors due to re-equilibration with target kinases after cell lysis and competition with covalent probe labeling. The estimated potency is affected by how inhibitors accumulate in cells, the dissociation rate constant (k_off_), and the duration of probe labeling. As a result, the cellular potency of HTH-01-091 is possibly greater than suggested by Figure 2. However, due to the lack of well-validated pharmacodynamic markers that can be used to specifically evaluate MELK inhibition, we cannot conclude that HTH-01-091 exhibits selectivity for MELK versus other intracellular targets. The complicating factors described above, in addition to high ATP concentrations inside cells, likely contribute to the substantial “cell shift” in the potency of the inhibitor.

2) The authors have not demonstrated that the dTAG-MELK fusion retains biological function. While it probably does, in the absence of a formal demonstration of that, it is not possible to use these findings to support a definitive conclusion about the putative lack of adaptive MELK independency based on the negative data that is presented.

Our ability to demonstrate that FKBP12^F36V^-MELK fusion protein retains the functionality of MELK is unfortunately limited by the absence of a well-validated and unique MELK substrate protein that could serve as a biomarker. As a surrogate, we evaluated what have been the most reliable phenotypes of MELK functionality, namely drug-induced degradation, and stabilization and hyperphosphorylation during mitosis, to demonstrate the relevance of FKBP12^F36V^-MELK.

We successfully demonstrated that FKBP12^F36V^-MELK recapitulates both phenotypes, which would suggest that this N-terminally tagged fusion MELK protein reasonably retains the endogenous binding partners and is similarly regulated (Figure 4—figure supplement 2). We have included the data in the second paragraph of the section that describes the dTAG system. However, we recognize that this does not definitely demonstrate FKBP12^F36V^-MELK functions. To account for this deficiency we have now added extensive new experiments where we address the consequence of MELK depletion using CRISPR-interference.

We tested 8 guide sequences around the transcription start site of MELK for their ability to knockdown MELK in MDA-MB-468 cells stably expressing KRAB-dCas9 (Figure 5). We also modified a doxycycline-inducible shRNA vector (tet-pLKO-puro) into an inducible sgRNA vector, so that we can compare the results of CRISPR interference with those of RNA interference (Figure 5). We confirmed by both qPCR and Western blotting that the five most effective guide RNAs worked as effectively as shMELK-1, 2 and 5 do after a 2-day doxycycline induction (Figure 5—figure supplement 2). Even though MELK knockdown with inducible CRISPRi is not as rapid as MELK degradation by the dTAG system (1-2 days vs. hours), we think both systems can adequately capture the immediate and transient response to loss of MELK, if it has an effect on cell proliferation. Based on the results of the five guide RNAs we tested, we found no evidence to support an adaptive MELK independency.

3) While it is great to see proteomic data confirming selective reductions of MELK in the dTAG degrader experiments, the actual magnitude of MELK reduction is very modest (~40%), which contrasts with the apparent complete loss of MELK expression observed by Western blotting (compare Figure 4). Presumably, this reflects that the proteomic data were acquired after only 1 hour of dTAG probe treatment. But, why was such an early time point selected for proteomic analysis of protein expression? It would have been arguably more interesting to see the proteomic profiles at 12 or 24 h post-dTAG probe treatment.

We originally selected a short time point of 1 hour because FKBP12^F36V^-MELK appears significantly degraded by this time point based on our Western blotting results. We agree that we may have learned more things about the system had longer time points been tested; however, the main goal of this experiment was to demonstrate selective FKBP12^F36V^-MELK depletion using an unbiased, proteome-wide approach. We worried that later time points would be confounded by secondary consequence of MELK protein depletion.

We also noticed the discrepant levels of MELK knockdown indicated by Western blotting versus by quantitative proteomics. In fact, in the first paper where the dTAG system was described (Erb et al., 2017), similarly, FKBP12^F36V^-ENL showed around 40% degradation in 3 hours (Erb et al., 2017, Figure 2) whereas a substantial loss of FKBP12^F36V^-ENL was already observed by an hour (Erb et al., 2017, Figure 2). However, we believe the Western blotting results are representative, and we offer a few possibilities of why quantitative proteomics may underestimate the extent of degradation.

First, the endogenous protein is knocked out because indels within the gene introduce premature stop codons. A shorter version of the protein or an exon-skipping isoform may still be translated, contributing to MELK-associated peptides. These potential MELK variants unlikely retain kinase activity because sgMELK-3 targets an exon within the kinase domain. Second, multiple peptide species may be present within the cells while FKBP12^F36V^-MELK is in the process of getting ubiquitinated and degraded by the proteosome. These modified and degraded fusion MELK proteins are unlikely functional, and are undetected by Western Blotting, but may still contribute to certain peptide populations detected by mass spectrometry. Last, five (four unique) MELK peptides identified in the multiplexed proteomic study were used to quantify the relative MELK protein levels collectively. When assessing each peptide individually (Table I), the relative abundance upon the degrader treatment ranged from 92% to 20%. This range of difference illustrates the limitation of quantitative multiplexed proteomics, and may support our hypothesis that certain background MELK peptides exist despite MELK knockdown and FKBP12^F36V^-MELK depletion. Unfortunately, our data do not support an obvious answer that the N-terminal peptides before the premature stop codon strictly contribute to this issue.

Table I. Five MELK peptides identified by quantitative multiplexed proteomics.

Raw MS3 ion intensityPeptide SequenceDMSODMSOdTAG-7dTAG-7dTAG-7Relative abundancePeptide positionK.TEIEALK.N286.15634238.127122160.527804153.643383163.87232161%N-terminal of premature stopK.TEIEALK.N36.270282237.103714234.585709934.9231331.698611192%N-terminal of premature stopK.VITVLTR.S133.405089115.32049948.257424652.175988259.892112443%C-terminal of premature stopK.LMTGVISPER.R59.635506548.517680215.633639923.633647516.148426834%C-terminal of premature stopK.NTLGSDLPR.I138.809172121.05887224.987950724.374933327.068258420%N-terminal of premature stop

4) In the bar graph in Figure 5, the individual bars should be labeled to show which shRNA is associated with each bar (as was done in panel 5B).

We have revised Figure 5 and fixed this problem.

5) The authors should cite a very recently published report in eLife (Lin et al., March 2017) that reaches similar conclusions regarding MELK's role in cancer cells.

We now mention and cite the study within the second paragraph of Discussion.

6) The title should be revised to something that is better matched to the more definitive conclusions that are described in the Abstract.

We have changed the title to “MELK is not necessary for the proliferation of basal-like breast cancer cells” to better describe what we learned from our study.